# General approach to surface-accessible plasmonic Pickering emulsions for SERS sensing and interfacial catalysis

Yingrui Zhang [1], Ziwei Ye [2], Chunchun Li[1], Qinglu Chen[1], Wafaa Aljuhani[1], Yiming Huang[1], Xin Xu [3], Chunfei Wu[1], Steven E. J. Bell [1] & Yikai Xu [1] ✉

Pickering emulsions represent an important class of functional materials with potential applications in sustainability and healthcare. Currently, the synthesis of Pickering emulsions relies heavily on the use of strongly adsorbing molecular modifiers to tune the surface chemistry of the nanoparticle constituents. This approach is inconvenient and potentially a dead-end for many applications since the adsorbed modifiers prevent interactions between the functional nanosurface and its surroundings. Here, we demonstrate a general modifier-free approach to construct Pickering emulsions by using a combination of stabilizer particles, which stabilize the emulsion droplet, and a second population of unmodified functional particles that sit alongside the stabilizers at the interface. Freeing Pickering emulsions from chemical modifiers unlocks their potential across a range of applications including plasmonic sensing and interfacial catalysis that have previously been challenging to achieve. More broadly, this strategy provides an approach to the development of surface-accessible nanomaterials with enhanced and/or additional properties from a wide range of nano-building blocks including organic nanocrystals, carbonaceous materials, metals and oxides.

Pickering emulsions consist of fine liquid droplets which are covered by a layer of solid particles and dispersed in another immiscible fluid. Pickering emulsions are traditionally stabilized with $SiO_2$ or polymer particles and have significant applications, for example in food manufacturing and cosmetics where they are used to create homogeneous dispersions with good long term stability[1,2]. More recently, the development of nanotechnology has led to a resurgence of Pickering emulsion-related research centered on the realization that the particle layer need not have a simple passive role and that the emulsions provide an accessible route into high surface area systems in which active particles sit at the interface between two immiscible liquids and are able to interact freely with the chemical compounds solubilized in both phases. This biphasic property gives Pickering emulsions immense potential in important emerging applications, such as controlled drug release or catalysis[3–8].

While some functional particles fortuitously also possess the appropriate surface chemistry needed to stabilize Pickering emulsions, the vast majority do not. The current approach to tailoring the surface chemistry of the particles is to use molecular "modifiers", which adsorb strongly to the surface of the functional material to neutralize surface charge and/or alter hydrophobicity, so that the particles can pack densely at the water-oil interface and stabilize the curved liquid meniscus[9–12]. While this approach can be effective, it is at best inconvenient, since it requires trail-and-error for each specific

[1]School of Chemistry and Chemical Engineering, Queen's University Belfast, University Road, Belfast BT7 1NN, UK. [2]Key Laboratory for Advanced Materials, Joint International Research Laboratory of Precision Chemistry and Molecular Engineering, Feringa Nobel Prize Scientist Joint Research Center, School of Chemistry and Molecular Engineering, East China University of Science & Technology, Shanghai 200237, PR China. [3]Collaborative Innovation Center of Chemistry for Energy Materials, Shanghai Key Laboratory of Molecular Catalysis and Innovative Materials, MOE Key Laboratory of Computational Physical Sciences, Department of Chemistry, Fudan University, Shanghai 200433, PR China. ✉e-mail: yxu18@qub.ac.uk

type of material and at worst may not be possible if a suitable modification method cannot be found. More importantly, this synthetic approach is potentially a dead end in many applications since the modifiers either prevent further interactions between the functional nanosurface and its surroundings, which is crucial for most applications[13,14], or they may be displaced in use, which leads to destabilization of the emulsion system. As a result, demonstrations of the potential applications of Pickering emulsions formed using modifiers have been very limited and are typically performed using carefully selected model systems.

The useful properties of Pickering emulsions can be partly exploited by introducing additional material processing procedures to transform them into colloidosomes or polymeric materials[15–19]. These procedures typically involve the removal of the dispersed phase to fix the position of the particles and thus form 3D assemblies. This makes it possible to replace the modifiers and access the active nanosurface without compromising the structure of the nanoparticle (NP)-assembly but loses the crucial advantages brought about by the biphasic nature of the parent emulsion. Alternatively, it is possible to form Pickering emulsions with particulate emulsion stabilizers which are chemically doped with a minuscule loading of functional materials on their surface[20–22]. This allows the biphasic nature and surface-accessibility of the Pickering emulsions to be preserved but offers little control over the morphology and, in turn, properties of the functional components.

In early work, Binks showed that the average wettability of the NP layer in Pickering emulsions stabilized by one type of $SiO_2$ nanoparticles ($SiO_2NPs$) can be adjusted by introducing a second type of $SiO_2NPs$ with an opposite wettability[23,24]. More recently, this concept was applied by Mao et al. to produce stable and photocatalytically active Pickering emulsions composed of $Ag_3PO_4$ and carbon nanotubes (CNTs)[25]. Inspired by the research above, here, we demonstrate a general and facile modifier-free approach that can be readily used to construct surface-accessible Pickering emulsions from nano- or microparticles with varying morphology, surface chemistry and material composition. The key to our approach lies in the combined use of "promoter" molecules and particulate "stabilizers", which remove interparticle electrostatic repulsion and provide emulsion stability, respectively, without passivating the surface of the functional materials. Depending on the surface properties of the particulate stabilizers, modifier-free o/w or w/o Pickering emulsions can be rationally obtained. Importantly, our approach leaves the surface of the functional materials free to interact with the surrounding environment without compromising the stability of the emulsions, which paves the way for realizing a range of important applications including biphasic catalysis and plasmonic sensing that had been previously impossible or highly challenging to achieve using modifier-based Pickering emulsions.

## Results

### Synthesis of modifier-free plasmonic emulsions

The process for synthesizing modifier-free plasmonic Pickering emulsions is simple. In practice, it only requires shaking a mixture of aqueous and/or oil colloids with a small amount of promoter (see Methods section for experimental details), as illustrated in Fig. 1a using CNTs and Au nanoparticles (AuNPs) as an example.

In general, the construction of a stable Pickering emulsion system involves, (1) the stabilization of NPs at the water-oil interface and (2) the stabilization of emulsion droplets using the NP layer. First, we analyze (1) by considering the adsorption energy, $\triangle G$, of a spherical

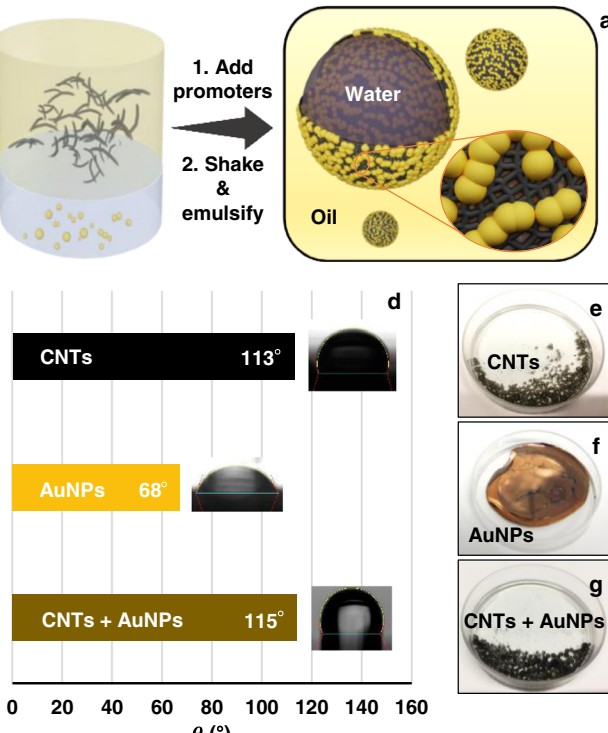

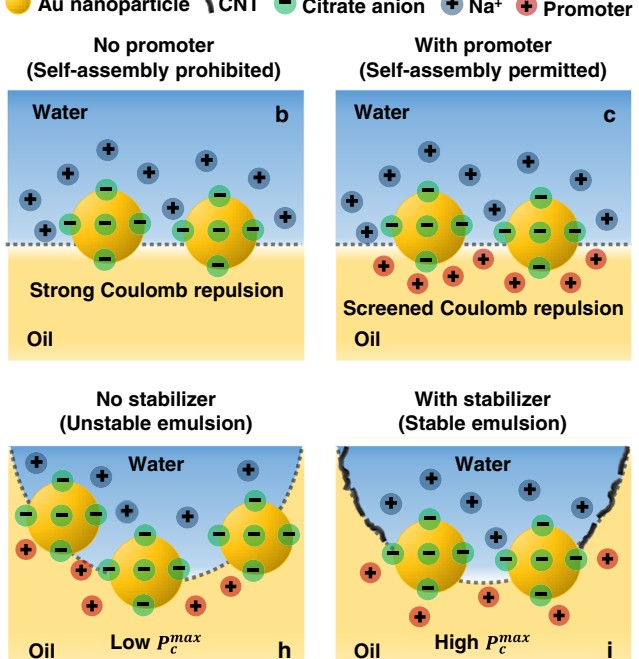

**Fig. 1 | Working principles of stabilizers and promoters in plasmonic Pickering emulsions. a** Schematic illustration of the experimental process for generating carbon nanotube (CNT)-Au nanoparticle (AuNP) stabilized w/o emulsions. **b** Schematic illustrations of charged AuNPs repelling each other at the interface. **c** Schematic illustrations of the working principle of organo-electrolytes acting as "promoters" for interfacial self-assembly by providing charge-screening. **d** Three-phase contact angle of CNTs, AuNP and CNT-AuNP layers measured for water-cyclohexane. **e–g** Photographs of CNT emulsions, AuNP films, and CNT-AuNP emulsions. **h** Schematic illustrations of unstable AuNP emulsions that can only withstand a low maximum capillary pressure ($P_c^{max}$). **i** Schematic illustrations of the working principle of CNTs acting as emulsion stabilizers by increasing $P_c^{max}$.

solid NP sitting at the interface between water and oil[23]:

$$\triangle G = -\pi R^2 \gamma_{wo}(1 \pm \cos\theta)^2 \qquad (1)$$

where $R$ is the radius of the NP, $\gamma_{wo}$ is the surface tension between the oil and water, $\theta$ is the three-phase contact angle measured through the dispersed phase on a solid particle in the environment of the dispersion medium and "+" and "−" signs refer to w/o emulsions and o/w emulsions, respectively.

Equation (1) shows that the adsorption of NPs to the water-oil interface is generally a favorable process, provided that $\theta$ is not 0° or 180°. Indeed, spontaneous interfacial self-assembly was observed for the uncharged CNTs dispersed in oil, when shaken with water. However, for electrostatically stabilized colloidal NPs, the overall change in system energy due to self-assembly can no longer be described using just Eq. (1), which is based solely on the reduction in interfacial energy. The additional gain in electrostatic potential from the NPs packing closely at the interface must be considered, which can be expressed as below[26].

$$\mu_{coulomb,w} = \frac{z\pi R^3 \sigma^2}{8\varepsilon_w \varepsilon_0}\left(3\sqrt{y}+\frac{2R}{L_D}\right)^{\frac{2R}{L_D\sqrt{y}}} \qquad (2)$$

$$\mu_{coulomb,o} = \frac{3z\pi R^3 \alpha^2 \sigma^2 \sqrt{y}}{8\varepsilon_{oil}\varepsilon_0} \qquad (3)$$

More specifically, Eqs. (2), (3) show the electrostatic potential between two similarly charged spherical NPs within a hexagonally packed NP-lattice on either side of the interface (Supplementary Fig. 1), where $\mu_{coulomb,w}$ and $\mu_{coulomb,o}$ are the electrostatic potential between the parts of the NPs immersed in the water and oil phases, respectively, $z$ is the number of nearest neighboring particles, $y$ is the fraction of interface covered by NPs versus the maximum fraction of interface that could be covered by the NPs, $\sigma$ is the particle charge density, $\varepsilon_{oil}$ and $\varepsilon_w$ are the dielectric constants of the oil and water phase, and $\varepsilon_0$ is the permittivity of vacuum. Since the charged species on the surface of the NPs are inherently hydrophilic, it is unlikely that they can be completely transferred across the interface into the oil phase. Additionally, some of the charged species which remain on the surface of the particles could recombine with their counter-ion upon entering the oil phase. Therefore, a scaling factor $\alpha$, which ranges between 0 and 1, is introduced in Eq. (3).

In practice, this means that electrostatically stabilized colloidal NPs, such as the AuNPs used here, do not migrate spontaneously to the interface when the colloid is simply shaken with an immiscible oil (Fig. 1b and Supplementary Fig. 2). Therefore, the key to inducing self-assembly of charged colloidal NPs at water-oil interfaces is to lower the electrostatic repulsion between adjacent particles at both sides of the interface. As discussed above, the conventional approach to achieve this is through the addition of organic modifiers, such as thiols, or co-solvents, such as ethanol, but modifiers passivate the surface of the functional NPs while co-solvents destabilize emulsion systems. Previously, we and others have shown that an alternative approach to induce self-assembly of charged colloidal NPs is to use promoters, such as tetrabutylammonium (TBA[+]) nitrate, which are lipophilic or amphiphilic organo-electrolytes that carry an opposite charge to the NPs[27–29]. As shown in Fig. 1c, the promoters along with the counterions, such as Na[+], which are initially present in the colloidal solution sit between the charged NPs at each side of the interface to provide charge-screening which helps overcome interparticle electrostatic repulsion. It is useful to note that this method is also fundamentally different from previous reports which use simple hydrophilic salts, such as NaCl, to induce partial aggregation of the colloid and therefore lower interparticle electrostatic repulsion at the interface[30]. Here, the

addition of promoters does not lead to any observable aggregation as shown by UV-vis spectroscopy (Supplementary Fig. 3). Therefore, using promoters allows the charged colloidal NPs to be assembled into densely packed superstructures without passivating their surface with strongly adsorbed modifiers or inducing undesired aggregation. In principle, the minimum amount of promoter required to effectively provide charge screening will differ depending on the surface-charge of the colloid used for self-assembly. However, this is not an issue in practice since a reasonable excess of promoter can be used routinely without perturbing the emulsion system. We have found that a promoter concentration of ca. $3 \times 10^{-5}$ M (per volume of colloid) was adequate for inducing self-assembly of highly charged colloids with zeta potential ca. −40 mV (Supplementary Fig. 4)[28]. Based on the above, a promoter concentration of $2.4 \times 10^{-3}$ M was consistently used for all emulsion systems in this work to mitigate interparticle electrostatic repulsion during self-assembly.

In addition to the successful assembly of NPs at the water-oil interface, the construction of stable Pickering emulsions also requires condition (2) to be met, which is that the particle layers at the interface need to be able to stabilize the thin liquid layer that is formed when two emulsion droplets move close to each other. This condition can be considered as a requirement for the NP bilayers to be able to withstand the pressing force between two approaching emulsion droplets. For densely packed solid spherical NPs, the maximum capillary pressure that can be withstood by the NP bilayer, $P_c^{max}$ can be defined as[31]:

$$P_c^{max} = \pm p \frac{2\gamma_{wo}}{R}(\cos\theta \pm z) \qquad (4)$$

with the emulsion system becoming more stable at higher $P_c^{max}$ values. In the equation, the sign "+" refers to o/w emulsions, and sign "−" refers to w/o emulsions, while $p$ and $z$ are parameters, which have positive values that increase with higher particle coverage at the interface (Supplementary Table S1).

Importantly, Eq. (4) shows that the value of $P_c^{max}$ increases as the NPs become more hydrophobic for w/o emulsions and more hydrophilic for o/w emulsions. As shown in Fig. 1d, $\theta_{Au}$ and $\theta_{CNT}$ were measured to be 68° and 113°, respectively. Although the wettability of both AuNPs and CNTs satisfy the minimum $\theta_{min}$ threshold of 50.7° required to stabilize w/o emulsions[31], the maximum pressing force that two CNT stabilized w/o emulsions could withstand was calculated to be ca. 5× higher than for the AuNP stabilized w/o emulsions (Supplementary Note 1). As a result, stable Pickering emulsions could only be obtained using CNTs, while emulsions formed with citrate-stabilized AuNPs coalesced in seconds into an interfacial film, as shown in Fig. 1e, f. Mixing the CNTs with AuNPs led to the formation of a mixed interfacial NP layer whose hydrophobicity resembled that of the plain CNTs rather than the AuNPs ($\theta_{CNT-Au} \approx \theta_{CNT} > \theta_{Au}$). This allowed the CNTs to act as emulsion stabilizers for constructing highly stable w/o Pickering emulsions that contained a layer of CNTs and AuNPs at the interface, as shown in Fig. 1g–i. It is worth noting that, in practice, the increase in the $P_c^{max}$ value with the addition of CNTs will be even higher than the value estimated above, since it has been shown that CNTs are able to cover a significantly higher proportion of surface area compared to the spherical geometry assumed for all particles in Eq. (4), which would lead to increased $p$ and $z$ values[32].

This approach is fundamentally different from typical examples of Pickering emulsions formed using binary components, since in those cases the binary components acted as co-stabilizers which combined to provide emulsion stability, while the Pickering emulsions shown here are stabilized solely by the CNT stabilizers[33–37]. Most importantly, the combined use of promoters and stabilizers removes the requirement to passivate the surface of the AuNPs with strongly adsorbing modifiers when constructing stable emulsions. The significant difference in the resulting surface chemistry of the plasmonic AuNPs is

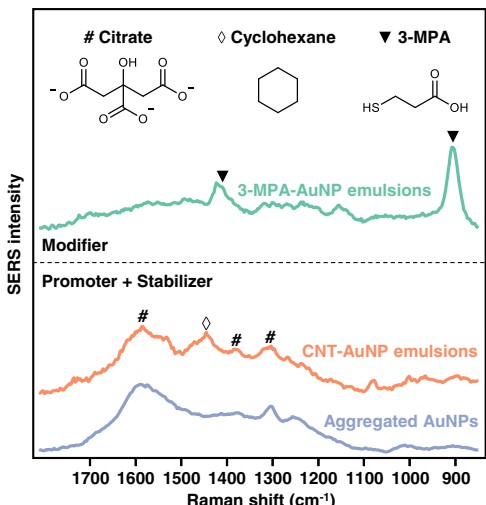

**Fig. 2 | Surface chemistry of modifier-capped and modifier-free plasmonic emulsions.** Surface-enhanced Raman spectroscopy (SERS) spectra obtained from 3-mercaptopropionic acid (3-MPA) modified AuNP emulsions (teal spectrum line), CNT-AuNP emulsions (rose spectrum line), and salt-aggregated AuNPs (ice blue spectrum line) in-situ. The spectra show that 3-MPA modifiers are adsorbed strongly to the AuNPs and alters their surface chemistry in the product emulsion while the surface chemistry of the AuNPs in emulsions formed with the combination of ionic promoters and particle stabilizers remain unchanged.

demonstrated using surface-enhanced Raman spectroscopy (SERS), which specifically enhances the Raman signature of the molecular species adsorbed on, or close to, plasmonic nanosurfaces[38]. As shown in Fig. 2, the SERS spectra of AuNP Pickering emulsions formed with 3-mercaptopropionic acid (3-MPA), which is a common type of modifier, are dominated by the molecular vibrations of 3-MPA adsorbed strongly on the AuNPs. In clear contrast, the SERS spectra of plasmonic Pickering emulsions formed using the promoter-assisted co-assembly approach resembled that of the parent citrate-capped Au colloid and showed only the molecular vibrations of the citrate and its derivates that were initially present on the NPs' surface as weakly adsorbed capping ligands.

The stability and average size of the Pickering emulsion droplets in relation to the water-oil ratio and the relative proportion of CNTs and AuNPs was explored. In general, it was found that the properties of the product emulsions were not highly sensitive to the water-oil ratio, provided that the volume of the dispersed phase remained unchanged (Supplementary Fig. 5). However, the size of the emulsion droplets was highly sensitive to the concentration of the CNT stabilizers. More specifically, as shown in Fig. 3a−f, the average size of the emulsion droplets was found to be inversely proportional to the concentration of CNTs used. This was expected, since increasing the number of CNTs would allow them to stabilize a larger interfacial area, which would, in turn, result in smaller emulsion droplets[17]. It can be seen from Fig. 3g and Supplementary Fig. 6 that the average size of the emulsion droplets formed from a mixture of CNTs and AuNPs was controlled solely by the concentration of the CNTs. More specifically, the average size of the CNT-AuNP emulsion droplets was found to be identical to those of the emulsions formed with just CNTs at the same CNT concentration, while altering the concentration of the AuNPs did not lead to obvious changes to the size of the emulsion droplets (Supplementary Fig. 7). This observation was consistent with the mechanism that the Pickering emulsions were stabilized solely by the CNT stabilizers proposed above. The fact that the addition of a substantial number of AuNPs to the water-oil interface did not generate any obvious changes to the size and morphology of the emulsions also suggested that there was already sufficient room at the water-oil interface of the CNT-stabilized

emulsions to accommodate the additional AuNPs. This assembly model is consistent with previous reports which showed that only a sub-monolayer coverage of the particulate stabilizers is required to form stable Pickering emulsions[39,40]. In contrast, the stability of the Pickering emulsions was found to be less sensitive to the change in the concentration of CNT stabilizers. As shown in Fig. 3g and Supplementary Fig. 8, even at the lowest CNT concentration, the product emulsions were stable for at least 7 days.

The arrangement of the binary NP layer was further studied using transmission electron microscopy (TEM) and scanning electron microscopy-energy dispersive X-ray spectroscopy (SEM-EDX) (see Methods section for details). Figure 3h−j show the TEM and SEM-EDX images of typical areas of the CNT-AuNP layer formed using 0.05 mg mL$^{-1}$ of CNTs. In general, it was found that regardless of the composition of the binary NP layer, the CNTs were always spread out evenly as an entangled mesh while the AuNPs occupied the gaps in the form of islands, which grew in size with increased concentrations of AuNPs in the Pickering emulsions (Supplementary Fig. 9). At the highest relative concentration of AuNPs to CNTs (corresponding to 0.025 mg mL$^{-1}$ of CNTs), the AuNP islands fused to form larger networks, at which point the stability of the corresponding Pickering emulsions decreased from >30 days to ca. 7 days (Fig. 3g). The arrangement of the particles at the interface was also probed using in-situ confocal SERS spectroscopy. As shown in Supplementary Fig. 10, a dramatic increase in the SERS signal intensity of the capping ligands adsorbed on the AuNPs was observed when the colloidal AuNPs were assembled from solution onto the surface of the emulsions. This suggested the formation of interparticle plasmonic hot spots during self-assembly[41], and is consistent with the ex-situ electron microscopy data shown above. The AuNPs preferred to form islands rather than mix evenly with the CNTs was likely due to the large differences in the surface chemistry of the two types of particles, and has been previously observed in other binary NP interfacial assemblies[42]. This particle arrangement is highly favorable for generating strong plasmonic properties, which underpin important applications in sensing, catalysis and photodynamic therapy[43−45].

## A platform technology for constructing Pickering emulsions

Since our promoter assisted co-assembly approach does not involve any chemically specific interactions, this means that the components within the self-assembly system are fully customizable and that the approach can serve as a platform technology for constructing various types of modifier-free Pickering emulsions. This is illustrated in Fig. 4a, which shows additional examples of "particle stabilizers", "functional materials" and "oils" that can be arbitrarily combined to generate Pickering emulsions with distinct properties. As shown in Fig. 4b−e, in addition to citrate reduced AuNPs, the functional materials can be readily changed, for example, to polyvinylpyrrolidone capped Au-Ag nanostars, polyvinylpyrrolidone capped Ag nanocubes, citrate capped polydisperse Ag nanoparticles (AgNPs) or even a combination of several different types of noble metal NPs. It is important to note that the functional materials above cannot form stable w/o Pickering emulsions without the addition of CNT stabilizers. In addition to CNTs, a diverse range of materials including pentacene nanocrystals and graphene nanopellets can also be used as hydrophobic stabilizers, as shown in Fig. 4f, g. Moreover, a variety of organic solvents, such as cyclohexane, dichloromethane, chloroform and toluene, can be used as the oil phase (Supplementary Fig. 11), which opens the possibility for introducing additional functional components, such as polymers, inorganic materials or functional molecules[17]. Finally, the same concept can also be readily expanded to the construction of o/w Pickering emulsions by simply changing the particulate stabilizers from hydrophobic to hydrophilic nanomaterials, such as $Cu_2O$, CuO, or $SiO_2$, as shown in Fig. 4h, i and Supplementary Fig. 12. Therefore, this approach potentially allows

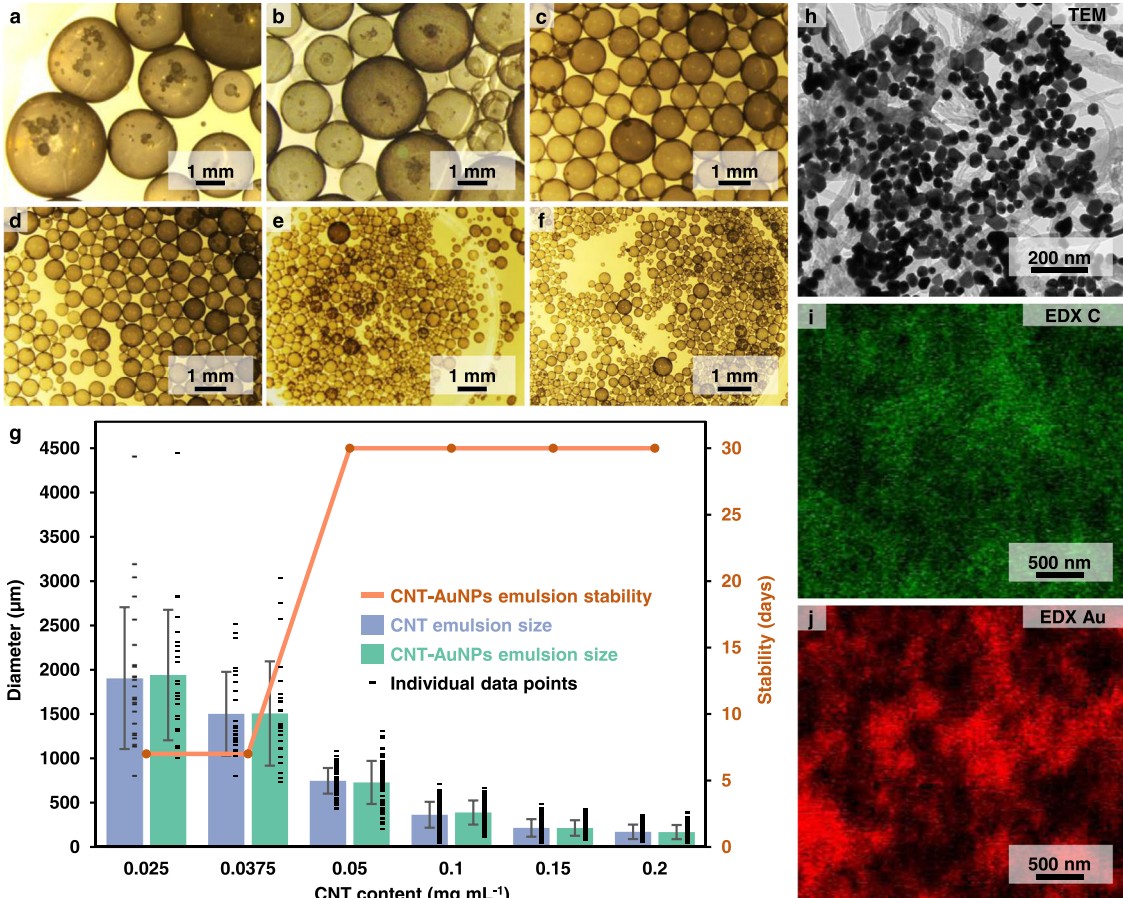

**Fig. 3 | Characterization of CNT-AuNP emulsions. a–f** Optical images of typical batches of CNT-AuNP emulsions formed with 0.025, 0.0375, 0.05, 0.1, 0.15, and 0.2 mg mL⁻¹ of CNTs, respectively. **g** Plot comparing the average size and stability of the emulsions versus the amount of CNTs used. The standard deviation for samples with a CNT content of 0.025 and 0.0375 were calculated from $N = 25$ individual emulsion droplets. The standard deviation for samples with a CNT content of 0.05 were calculated from $N = 80$ individual emulsion droplets. The standard deviation for samples with a CNT content of 0.15 and 0.2 were calculated from $N = 100$ individual emulsion droplets. **h** Transmission microscopy image of a typical area in a CNT-AuNP layer formed using 0.05 mg mL⁻¹ of CNTs. **i–j** Energy dispersive X-ray spectroscopy mapping of the C (green) and Au (red) content of a typical area in a CNT-AuNP layer formed using 0.05 mg mL⁻¹ of CNTs. The data shown in **h** and **i**, **j** were obtained from different samples.

various composite materials with synergistic functionalities to be rationally constructed.

**Biphasic plasmonic sensing with w/o Pickering emulsions**
The ability to produce surface-accessible Pickering emulsions, which remain highly stable despite changes to the surface chemistry of its NP constituents, enables the development of important applications that require direct access to the functional nanosurface, such as plasmonic sensing and catalysis. Here, the plasmonic properties of the CNT-AuNP Pickering emulsions were investigated using SERS with crystal violet as the model analyte (see Methods section and Supplementary Fig. 13 for experimental details). By analyzing the SERS signal intensity of the characteristic peak of crystal violet ($10^{-5}$ M) at 1173 cm⁻¹, it was found that the SERS enhancement from the emulsions decreased linearly as the concentration of CNTs increased (Fig. 5a, b). This was unsurprising since, as discussed above, an increased CNT concentration led to a larger interfacial area, and in turn a lower density of AuNPs on the surface of each emulsion droplet. Although the Pickering emulsions formed with 0.025 mg mL⁻¹ of CNTs generated the strongest SERS enhancement, they were considerably less stable than emulsions formed with higher concentrations of CNTs, as discussed above. Therefore, Pickering emulsions formed with 0.05 mg mL⁻¹ of CNTs were selected as the optimal plasmonic substrate for the applications below.

The concept of plasmonic emulsions was first demonstrated more than 30 years ago[46], and have since been proposed as strong enhancing substrates for SERS[3,9,10]. However, this potential application has barely been fulfilled since the strongly-adsorbed modifiers, which had been essential for constructing the plasmonic emulsions, prevent analyte molecules from interacting with the plasmonic nanosurface and accessing the enhancing hot spots (illustrated in the schematic of Fig. 5c). For example, Fig. 5c compares the SERS performance of AuNP emulsions formed using 3-MPA modifiers, against AuNP emulsions formed with promoters and CNT stabilizers in the detection of adenine. Even though adenine is a biomolecule which is known to adsorb spontaneously to the surface of AuNPs, the adenine molecules were unable to displace the strongly adsorbed 3-MPA modifiers. As a result, no signals of the analyte could be observed from the modifier-capped emulsions when they were allowed to interact with $10^{-4}$ M of adenine (Fig. 5c). In sharp contrast, the surface of the AuNPs in the CNT-AuNP emulsions only contained weakly adsorbed citrate and chloride capping ligands, which could be easily displaced by adenine (Supplementary Fig. 14). As a result, clear SERS signals of adenine could be obtained even when its concentration was 1000× lower ($10^{-7}$ M) than the concentration used for modifier-capped emulsions (Fig. 5d). Additional data on the performance of the CNT-AuNP emulsions as enhancing substrates for quantitative SERS is shown in Supplementary Fig. 15. Briefly, even with simple citrate-reduced AuNPs as the

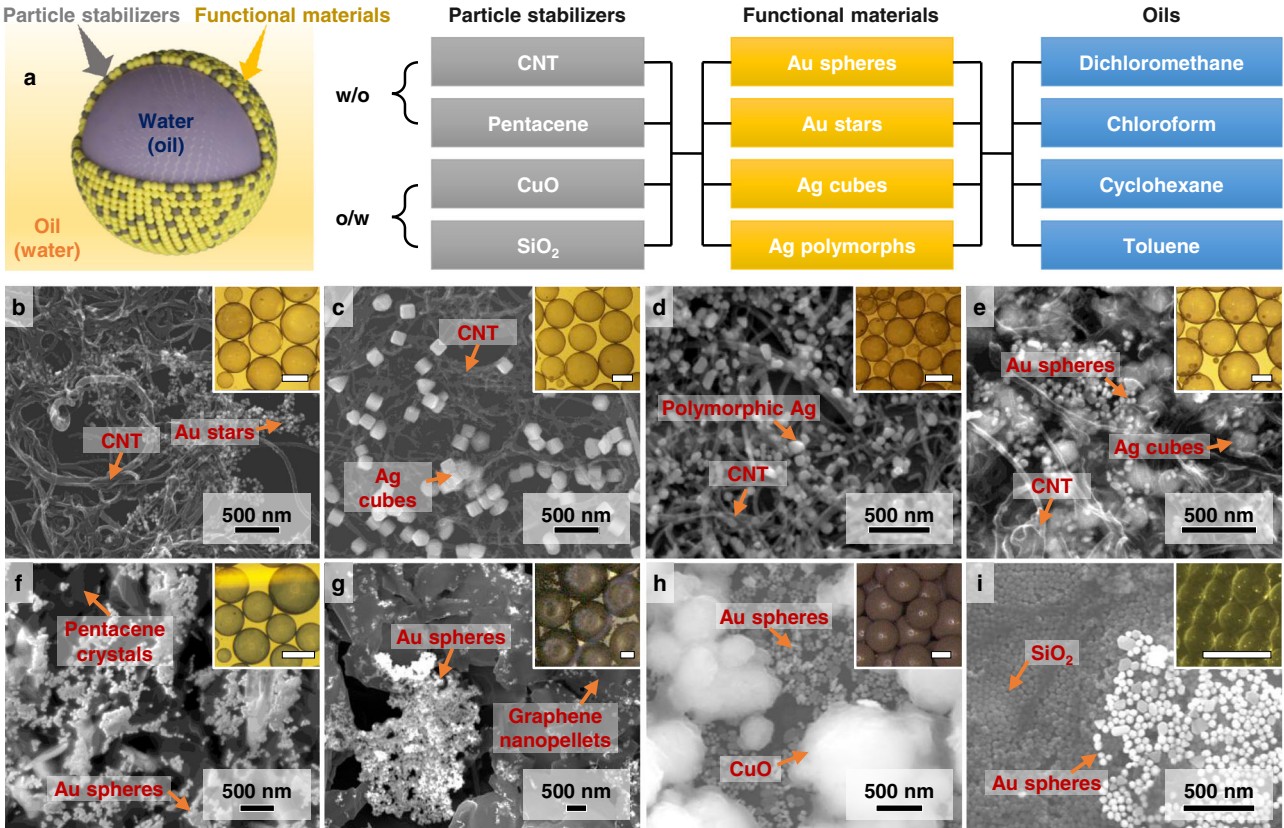

**Fig. 4 | Expanding Pickering emulsion types via promoter assisted co-assembly.** **a** Schematic illustrations showing the key interchangeable components within the promoter assisted co-assembly of Pickering emulsions. **b**–**i** SEM of the mixed NP layer in emulsions stabilized with CNTs and Au nanostars; CNTs and Ag nanocubes; CNTs and polymorphic Ag nanocrystals; CNTs, Au nanospheres and Ag nanocubes; pentacene crystals and Au nanospheres; graphene nanopellets and Au nanospheres; CuO microparticles and Au nanospheres; SiO₂ and Au nanospheres. Insets show the optical images of the emulsion samples. The scale bars in the optical images correspond to 500 μm.

plasmonic component, the limits of detection for adenine and crystal violet could reach $10^{-7}$ M and $5 \times 10^{-7}$ M, respectively. The linear quantitation range was determined to be from $10^{-4}$-$10^{-7}$ M and $10^{-5}$-$5 \times 10^{-7}$ M for adenine and crystal violet, respectively. It is important to note that the significant improvement in SERS performance observed for CNT-AuNP emulsions over modifier-capped emulsions is not specific to adenine. Figure 5e–g and Supplementary Fig. 16 show the SERS signals of a range of important analyte molecules including drugs (nicotine and panobinostat)[47], pesticides (thiram)[48], environmental pollutants (aniline)[49] and food contaminants (naphthalene, biphenyl, pyrene)[50] which can be obtained using CNT-AuNP but not 3-MPA-capped emulsions.

A unique advantage that comes with using the modifier-free plasmonic emulsions as SERS substrates is their biphasic nature, which allows the construction of dual-phase sensors. More specifically, since the plasmonic NPs sit at the water-oil interface, this allows analyte molecules to be introduced from both liquid phases, which significantly adds to the versatility of the substrate and is crucial for the detection of non-water-soluble analytes, such as polyaromatic hydrocarbons (Fig. 5f and Supplementary Fig. 16). Moreover, a diverse range of organic solvents can be used to suit the application. For example, cyclohexane was used as the oil phase in our typical SERS measurements, but *n*-dodecane was used for SERS detection of nicotine and panobinostat to avoid interference between the Raman background of cyclohexane and the SERS signals of the analytes (Fig. 5g and Supplementary Fig. 17). The biphasic properties of the emulsions can also be exploited for the simultaneous detection of analyte molecules with different hydrophobicity, which has important potential applications in studies, such as in-situ reaction monitoring or environmental

analysis[51,52]. As proof-of-principle, Fig. 5h and Supplementary Fig. 18 demonstrate the simultaneous detection of adenine and naphthalene adsorbed from the aqueous and oil phase, respectively. As shown in the spectrum, the vibrational fingerprints for both types of molecules could be clearly observed and distinguished, which demonstrates the feasibility of the approach.

The fact that the plasmonic NPs are no longer relied upon for providing emulsion stability also means that their surface can be tailored to introduce advanced functionalities without affecting the stability of the emulsions. To illustrate this, the surface of the AuNPs was functionalized with a self-assembled monolayer of 4-mercaptobenzoic acid (4-MBA). Similar to the citrate-capped AuNPs, 4-MBA-capped AuNPs cannot stabilize Pickering emulsions alone but could form highly stable emulsions when co-assembled with CNT stabilizers. As shown in Supplementary Fig. 19, the SERS spectra of 4-MBA showed distinct vibrational bands at different pH values depending on the protonation state of their carboxylic functional groups, which gives the 4-MBA-functionalized plasmonic emulsions the potential to act as pH sensors[53].

## Interfacial catalysis with o/w Pickering emulsions

It is now well-established that strongly adsorbed chemical ligands have a negative impact on the catalytic activity of nanomaterials, since catalytic reactions can only be initiated when the reactants have access to the catalytic surface[54]. This creates a particularly challenging dilemma with traditional emulsion-based nano-catalyst systems, in which emulsion stability and catalytic activity must be carefully balanced. The fact that our current method uncouples the need for the functional particles to provide both stabilization and functionality

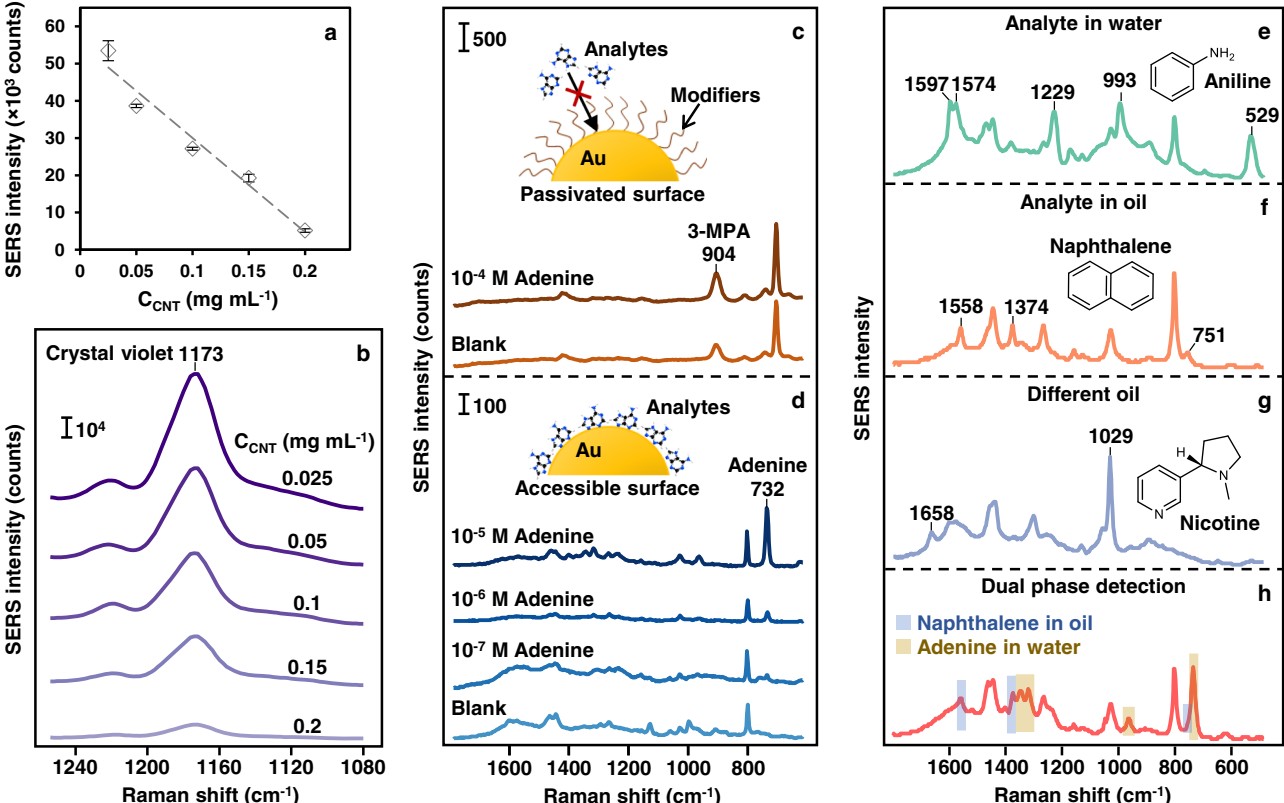

**Fig. 5 | SERS sensing with w/o CNT-AuNP Pickering emulsions. a** Plot showing the SERS signal intensity of crystal violet at $10^{-5}$ M obtained using CNT-AuNPs emulsions prepared at different CNT concentrations. The SERS intensity of crystal violet was calculated by measuring its characteristic peak at 1173 cm⁻¹. The standard deviations were calculated from $N = 3$ independent samples. **b** The SERS spectra of crystal violet corresponding to the data points in **a**. **c** SERS spectra obtained from 3-MPA-capped AuNP emulsions before and after being treated with $10^{-4}$ M of adenine. No signals of adenine can be observed since the surface of the AuNPs is blocked by the 3-MPA modifiers. **d** SERS spectra obtained from CNT-AuNP emulsions treated with various concentrations of adenine. The surface of the AuNPs is accessible which allows clear signals of adenine to be observed. **e** SERS spectra of aniline at $10^{-4}$ M obtained on CNT-AuNP emulsions as an example for the detection of analytes in water. **f** SERS spectra of naphthalene at $10^{-4}$ M obtained on CNT-AuNP emulsions as an example for the detection of analytes in oil. **g** SERS spectra of nicotine at $10^{-4}$ M obtained on CNT-AuNP emulsions formed with *n*-dodecane instead of cyclohexane as an example for the detection of analytes in different oils. **h** SERS spectra of adenine at $10^{-6}$ M and naphthalene at $10^{-4}$ M obtained on CNT-AuNP emulsions as an example for the simultaneous detection of analytes in water and oil. The SERS intensity of the spectra presented in **e–h** have been normalized for illustration purposes.

provides a solution to this long-standing challenge. As shown in Fig. 6a and Supplementary Fig. 20, the catalytic performance of the modifier-free Pickering emulsions was compared with their modifier-capped counterparts using the reduction of 4-nitrophenol with sodium borohydride and o/w SiO₂NP-noble metal NP emulsion catalysts as the example.

As shown in Fig. 6b, since the H⁻ ions present in NaBH₄ solutions adsorb strongly to the surface of noble metal NPs and are able to replace even thiols[54,55], the addition of NaBH₄ solution led to the desorption of the thiol modifiers, which immediately destabilized the modifier-capped emulsions. As a result, the emulsions coalesced into several large liquid droplets within minutes. In sharp contrast, since the SiO₂ stabilizers are chemically inert, the SiO₂NP-metal NP Pickering emulsions remained completely stable on addition of NaBH₄ solution and throughout the catalytic reaction (Fig. 6c and Supplementary Fig. 21). As shown in Fig. 6d, e, using SiO₂NP-AgNP Pickering emulsions formed with as-prepared citrate-reduced Ag colloid as the functional catalyst, the catalytic reduction of 4-nitrophenol with NaBH₄ followed a pseudo first order with a rate constant ($k_{Ag}$) of ca. $2.58 \times 10^{-3}$ s⁻¹ and a conversion rate of 92% (see Supplementary Fig. 22 and Supplementary Note 2 for details). As discussed in Supplementary Figs. 4 and 10, since a negligible amount of metal NPs remain in the aqueous phase after emulsion formation, the catalytic activity of the emulsions can be attributed solely to the metal NPs at the water-oil interface. Table 1 compares the catalytic

performance of the unoptimized SiO₂NP-AgNP Pickering emulsions with other state-of-the-art emulsion-based interfacial catalysts designed specifically for the reduction of 4-nitrophenol using NaBH₄[56–60]. In general, even though the SiO₂NP-AgNP Pickering emulsions were unoptimized and contained ca. 6-19× less metal catalyst, they showed comparable or faster reaction rates compared to the modified emulsions constructed using carefully optimized nanostructures and metal compositions. In addition to the superior catalytic activity, the modifier free emulsions also showed excellent stability over time. As shown in Fig. 6f and Supplementary Fig. 23, the SiO₂NP-AgNP emulsions fully retained their catalytic activity even when left at room temperature for 1 month. Finally, the generality of our co-assembly approach and stability of the product emulsions means that the composition of the emulsions and the reaction conditions can be readily adjusted to suit specific catalytic reactions. For example, Supplementary Figs. 24–27 shows the catalytic performance of the modifier-free emulsions tested at different reaction temperatures, reactant concentrations, stabilizer concentrations and catalyst composition. Consistent with literature, it was found that the rate constant and conversion rate of the reaction was highly dependent on the experimental parameters mentioned above. For example, $k_{Ag}$ increased to ca. $4.24 \times 10^{-3}$ s⁻¹ when the reaction temperature was simply increased from 15° to 35°, which shows the potential of our approach in the construction of highly active biphasic catalytic systems.

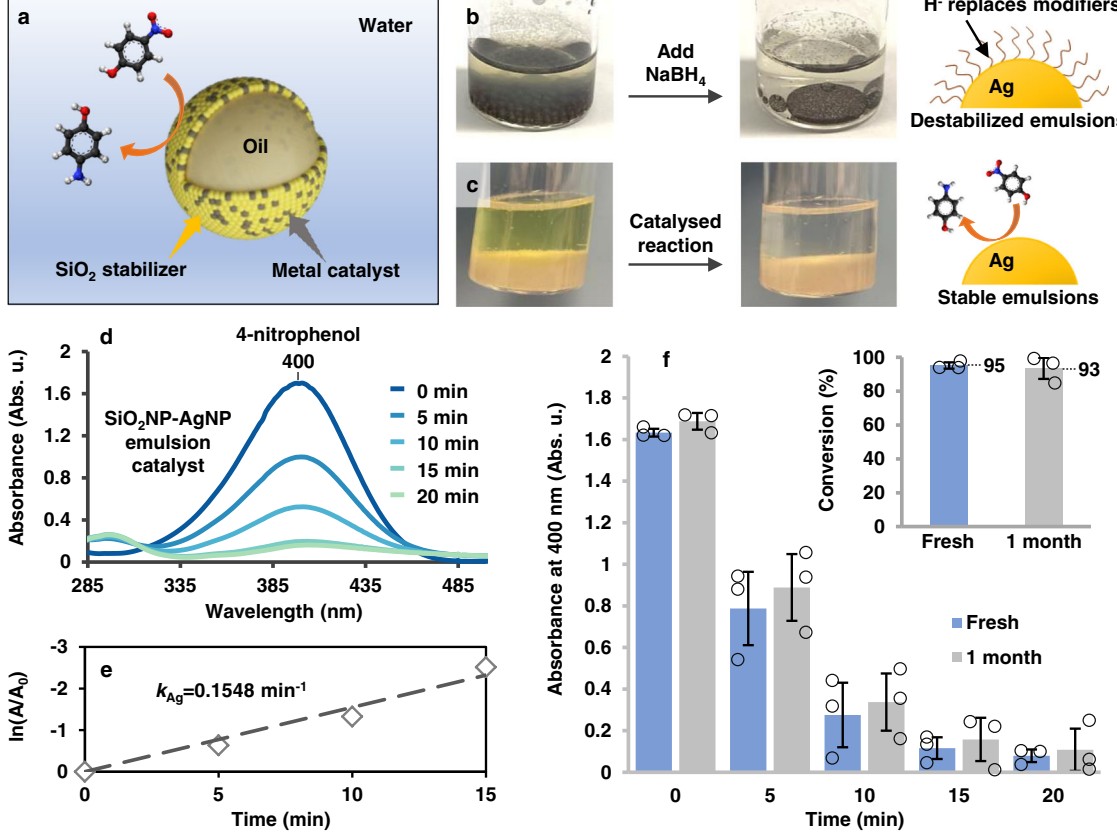

**Fig. 6 | Interfacial catalysis with o/w SiO₂NP-AgNP Pickering emulsions.**
**a** Schematic illustrations of a modifier-free o/w SiO₂NP-metal NP emulsion droplet acting as an interfacial catalyst for the reduction of 4-nitrophenol to 4-aminophenol. **b** Optical images of a Ag emulsion sample formed with 3-MPA modifiers before and a few minutes after being treated with NaBH₄ solution. Also shown is the schematic illustrations of the 3-MPA modifiers being replaced by H⁻ in the NaBH₄ solution leading to emulsion destabilization. **c** Optical images of a SiO₂NP-AgNP emulsion sample before and after 20 min of catalytic reaction. Also shown is the schematic illustrations of the surface-accessible AgNPs in modifier-free emulsions acting as stable catalysts. **d** UV−vis spectra of 4-nitrophenol

measured at different stages of its catalytic reduction using SiO₂NP-AgNP Pickering emulsions as the catalyst. **e** Semi-log plot of $\ln(A/A_0)$ against time for 4-nitrophenol. The absorbance is measured for the peak at 400 nm using the data shown in **d**. **f** Plot showing the absorbance of 4-nitrophenol measured by UV−vis spectroscopy at different points of catalytic reduction using fresh SiO₂NP-AgNP Pickering emulsions and SiO₂NP-AgNP Pickering emulsions that have been stored for 1 month as the catalyst. Inset compares the overall conversion of 4-nitrophenol using fresh and stored SiO₂NP-AgNP emulsions. The standard deviations in both charts were calculated from $N = 3$ independent samples.

## Table 1 | Emulsion-based interfacial catalysis

| Emulsion-based catalyst | Material (wt%) | Catalyst (wt%) | $n$ (mmol) | $C$ (mM) | $T$ (min) | $k$ (s⁻¹) | CVR (%) | Ref. |
|---|---|---|---|---|---|---|---|---|
| AgNP-JMP | 1.3 | $2.6 \times 10^{-2}$ | $2.25 \times 10^{-4}$ | / | 30 | / | 97 | [56] |
| ANPL$_{Ag}$ | 0.25 | $1.8 \times 10^{-2}$ | $2.25 \times 10^{-4}$ | / | 4 | / | 100 | [57] |
| m-MOP/Ag@polyHIPE | 1 | $1.6 \times 10^{-2}$ | $5 \times 10^{-4}$ | 0.05 | 40 | $6.17 \times 10^{-4}$ | / | [58] |
| J-AuNPs | 0.02 | $8.8 \times 10^{-3}$ | $8 \times 10^{-5}$ | 0.1 | 10 | / | 100 | [59] |
| Ni-HPF | 1 | / | $5 \times 10^{-3}$ | 0.1 | 24 | $2.58 \times 10^{-3}$ | >90 | [60] |
| SiO₂NP-AgNP | 0.063 | $1.4 \times 10^{-3}$ | $8 \times 10^{-4}$ | 0.1 | 15 | $2.58 \times 10^{-3}$ | 92 | This work |

Comparison of catalytic performance and experimental parameters of Ag NPs on amphiphilic Janus microparticles (AgNP-JMP)[56], amphiphilic nanoplatelets (ANPL) modified with Ag NPs[57], Ag-incorporated microporous organic polymer (m-MOP/Ag) particles stabilized high internal phase emulsion (HIPE)[58], Janus particles modified with AuNPs (J-AuNPs)[59], Nickel-coated hyperporous polymer foam (Ni-HPF)[60], and SiO₂NP-AgNP Pickering emulsions. The material content (Material) is defined as the weight percentage of the whole material in the aqueous catalytic system. The catalyst content (Catalyst) is defined as the weight percentage of the catalytically active material in the aqueous catalytic system. Quantities that were undetermined are labeled as "/". The abbreviations "$n$", "$C$", "$T$", "$k$" and "CVR" corresponds to the amount of 4-nitrophenol, the concentration of 4-nitrophenol, the reaction time, the rate constant and the conversion rate, respectively.

## Discussion

In conclusion, we have demonstrated a strategy for constructing stable and modifier-free Pickering emulsions. This was achieved using a promoter assisted co-assembly approach in which the functional NPs were assembled along with another type of NP that acted as the emulsion stabilizer. By altering the concentration of the stabilizer particles, the emulsion size, stability and functional particle coverage

can be rationally controlled. We showed that this approach provides a general platform for synthesizing Pickering emulsions carrying nano- or micro-particles of different morphologies, compositions, surface chemistry, and in turn, functionalities. Depending on the hydrophobicity of the particle stabilizers, w/o and o/w emulsions can be made. Importantly, this approach provides a step change from the current method since it removes the need for chemical modifiers and

in turn resolves the long-standing dilemma of choosing between stability and functionality in the synthesis of Pickering emulsions. This opens the door for various important applications that have been envisaged for Pickering emulsions but were challenging to realize, such as biphasic sensing and catalysis. This was demonstrated using Pickering emulsions formed from noble metal functional NPs and CNT/SiO$_2$ particulate stabilizers. The fact that the surface properties of the functional NPs are no longer tied to the stability of the emulsions also allows the surface of the functional NPs to be tailored to introduce advanced functionalities, such as pH recognition. Importantly, since our promoter-assisted co-assembly approach can be extended to other nanomaterials, this paves the way for the better utilization of the wide range of rationally designed functional NPs available in literature for the construction of modifier-free Pickering emulsions with enhanced and/or additional functionalities. More broadly, we anticipate that the results of this work will spark new research interest in the development of surface-accessible nanomaterials, which are not limited to Pickering emulsions but also include NPs, nano-assemblies and nano-hybrid materials.

## Methods

### Materials
Panobinostat was purchased from Selleck Chemicals. CuO and Cu$_2$O microparticle powders were purchased from Alfa Aesar. Silica nanospheres (50 nm, 10 mg mL$^{-1}$) were purchased from nanoComposix. Graphene nanopellets were purchased from ACS Material®. Other chemicals were purchased from Sigma-Aldrich®. All chemicals were used as received. Distilled deionized (DDI) water with a low resistivity of 18.2 MΩ cm was used for all experiments.

### Colloid synthesis
Citrate-reduced AuNPs were synthesized based on the Turkevich-Frens method with slight modifications[61]. Briefly, 0.05 g of HAuCl$_4$·3H$_2$O was dissolved in 50 mL of DDI water and heated with vigorous stirring under reflux until boiling. Then, 5.6 mL of aqueous trisodium citrate dihydrate (1 wt%) was added to the precursor solution all at once. The boiling colloid was left to age under stirring for 15 min before being cooled down naturally to room temperature. The product colloid could be stored for at least 3 months at 4 °C and was used directly without any additional treatment.

Monodisperse citrate-reduced AuNPs were prepared following the method by ref. [62]. Briefly, 150 mL of 2.2 mM trisodium citrate dihydrate solution was first heated to boil under reflux and magnetic stirring, then 1 mL of 25 mM HAuCl$_4$·3H$_2$O solution was added all at once. After this, the colloid was boiled for 15 min leading to the formation of Au seeds. Immediately after this, and in the same reaction vessel, the temperature of the seed solution was cooled down to 90 °C before 1 mL of 25 mM HAuCl$_4$ solution was injected into the solution and allowed to react for 30 min under reflux and stirring. This process was repeated once more before being allowed to cool to room temperature. The resulting AuNPs were ca. 20 nm in diameter, could be stored for at least 3 months at 4 °C and were used directly without any additional treatment.

Polydisperse citrate-reduced AgNPs were synthesized based on the Lee and Meisel method with slight modifications[63]. Briefly, 45 mg of AgNO$_3$ was dissolved in 250 mL of DDI water and heated up to boil under reflux with vigorous stirring. After this, 5 mL of 1 wt% trisodium citrate dihydrate (aq.) was added to the boiling solution continuously within 30 s with a syringe. Then the solution was allowed to boil for 90 min before being allowed to cool to room temperature. The product colloid could be stored for at least 3 months at 4 °C and was used directly without any additional treatment.

Monodisperse citrate-capped Pt nanoparticles (PtNPs) were prepared based on the method by Kim et al. with slight modifications[64]. Firstly, 3 mL of 0.2 wt% H$_2$PtCl$_6$·6H$_2$O solution was added to 39 mL of

boiling DDI water under reflux and stirring. After 1 min, 0.92 mL of 1 wt% trisodium citrate solution was added to the above mixture. After another 30 s, a 0.46 mL of solution containing 0.08 wt% NaBH$_4$ and 1 wt% sodium citrate was added into the reaction solution. The boiling solution was allowed to react under stirring for another 10 min before being cooled down naturally to room temperature. This seed solution could be stored for at least 3 months at 4 °C. To grow the seeds into larger PtNPs, 1 mL of the seed solution, 4.5 mL of 4 mM H$_2$PtCl$_6$·6H$_2$O solution, and 0.5 mL of solution containing 1.25 wt% L-ascorbic acid and 1 wt% trisodium citrate dihydrate were added sequentially into 25.5 mL of DDI water. The solution was brought to boil and allowed to react for 30 min, before being allowed to cool to room temperature. This led to the formation of PtNPs ca. 28 nm in diameter, which could be stored for at least 3 months at 4 °C and used directly without any additional treatment.

Au nanostars were synthesized strictly following a protocol by ref. [65]. Immediately after synthesis, 1 mL of 0.1 wt% PVP (aq.) was introduced to the colloid and stirred for 5 min to form stable PVP-nanostar colloids, which remained stable for at least 1 month at room temperature. PVP-Ag nanocubes were synthesized strictly following a protocol by ref. [66].

### Synthesis of w/o Pickering emulsions through promoter assisted co-assembly
In general, CNT stabilized w/o emulsions were formed by shaking a mixture of CNTs dispersed in oil with aqueous colloid and aqueous promoter solution (oil-water ratio 4:1). First, CNT powder was dispersed into an oil (cyclohexane, $n$-dodecane, dichloromethane, toluene, chloroform), through ultra-sonication. This was performed by adding 2 mg of CNT powder into 20 mL of cyclohexane and sonicating using a Soniprep 150 Ultrasonic Disintegrator with 65 watts power for 30 min. 1 (2) mL of this CNT cyclohexane solution was extracted and diluted by 2× with cyclohexane to be used as the oil phase for self-assembly. The diluted CNT cyclohexane solution was shaken vigorously with 500 μL of citrate-reduced Au (PVP-capped Au nanostars; PVP-capped Ag nanocubes) colloid and 50 μL of 1 mM TBA$^+$NO$_3^-$ (aq.) for 30 s, which led to the formation of Pickering emulsions. To form Pickering emulsions with CNT and citrate-reduced Ag colloids, 2 mL of CNT cyclohexane solution was extracted and diluted by 2× with cyclohexane to be used as the oil phase, which was shaken with 1 mL of colloid and 100 μL of 1 mM TBA$^+$NO$_3^-$ (aq.) for self-assembly. To synthesize w/o Pickering emulsions with graphene nanopellets (pentacene), 10 (0.5) mg of graphene nanopellet (pentacene crystal) powder was dispersed into 10 (1) mL of dichloromethane. After this, 4 (1) mL of the graphene nanopellet (pentacene) dichloromethane solution was shaken vigorously with 0.5 (0.3) mL of citrate-reduced Au colloid and 100 (50) μL of 1 mM TBA$^+$ NO$_3^-$ (aq.) for 30 s, which led to the formation of Pickering emulsions.

### Synthesis of o/w Pickering emulsions through promoter assisted co-assembly
To synthesize o/w Pickering emulsions with CuO (Cu$_2$O), 1 (0.05) g of CuO (Cu$_2$O) powder was dissolved in 5 mL of DDI water and sonicated for 5 min. Citrate reduced Au colloid was concentrated by 10× by centrifuging at 372 × $g$ for 30 min and then redispersing the colloid in the appropriate lower volume of water. After this, 5 mL of the CuO (Cu$_2$O) solution was shaken with 0.3 mL of the concentrated citrate-reduced Au colloid, 3 mL of dichloromethane, and 140 μL of 1 mM TBA$^+$NO$_3^-$ (aq.) for 30 s, which led to the formation of Pickering emulsions. The synthesis of SiO$_2$NP-metal NP Pickering emulsions is discussed below.

### Synthesis of Pickering emulsions with modifiers
Modifier-capped Pickering emulsions were prepared according to our previous work[17]. Briefly, Pickering emulsions were synthesized by

vigorously shaking 3 mL of citrate-reduced Au (Ag) colloid with 1 mL of dichloromethane, 100 μL of 1 mM $TBA^+NO_3^-$ (aq.) and 100 μL of 10 mM 3-MPA (aq.) for 30 s, which led to the formation of 3-MPA-capped AuNP Pickering emulsions.

## SERS using CNT-AuNP Pickering emulsion

SERS detection of crystal violet, thiram, aniline, naphthalene, biphenyl, pyrene, nicotine and panobinostat was performed using a Perkin Elmer RamanMicro 200 Raman Microscope with an excitation wavelength of 785 nm (10× lens, 15 mW laser power, 20 s accumulation time). In other cases, SERS detection was performed using a WITec Alpha 300 R Confocal Raman Microscope equipped with a 785 nm laser (10× lens, 60 mW laser power, 30 s accumulation time). All Pickering emulsions were freshly prepared and used immediately. SERS samples were prepared by adding 50 μL of aqueous analytes (or 20 μL of oil dispersed analytes) into the emulsions and then shaking the sample to facilitate adsorption. For SERS analysis, an appropriate amount of Pickering emulsion was poured into a hydrophobic polymer container, and the laser was focused onto the surface of a Pickering emulsion droplet to obtain SERS signals. Since the analytes are adsorbed evenly onto the NPs in each Pickering emulsion droplet and only a single area on one emulsion droplet is analyzed each time, this means that there is no need to control the exact amount of emulsion poured from the parent sample for each analysis. All spectra presented in the paper were averaged from at least from 3 independent measurements using GRAMs AI.

4-MBA functionalized Au colloids were prepared by adding 100 μL of $10^{-4}$ M 4-MBA (aq.) to 1 mL of citrate-reduced Au colloid and then incubated for 20 min to facilitate adsorption. The pH value of the 4-MBA capped AuNPs was adjusted to 3 and 7 by adding an appropriate amount of 0.1 M NaOH solution. The 4-MBA capped AuNPs were used to synthesize Pickering emulsions using CNT stabilizers and cyclohexane as the oil phase via the exact same procedure as used for the citrate-reduced AuNPs. The SERS of the 4-MBA capped CNT-AuNP Pickering emulsions were obtained using WITech Alpha 300R confocal Raman microscope, as described above.

## Microscopy characterizations

SEM-EDX was performed using a Quanta FEG 250 at an acceleration voltage of 30 kV under high vacuum ($8 \times 10^{-5}$ mbar). The SEM samples were prepared by adding 100–200 μL of ethanol into the Pickering emulsion samples to destabilize the emulsions and form interfacial arrays which were dried onto aluminum foil in a vacuum oven. Optical microscopy of the Pickering emulsions was performed using a Nikon SMZ800 microscope. TEM images were recorded using a Joel JEM-1400 Plus Transmission Electron Microscope at an acceleration voltage 80 kV. The TEM samples were fabricated by extracting a Pickering emulsion droplet with a carbon film covered TEM grid (S160,200 mesh Cu (25)), then drying at room temperature.

## Contact angle measurements

Contact angle measurements were conducted using a First Ten Angstroms FTA1000 goniometer. The samples were fabricated in the same way as the SEM samples. The samples were mounted at the bottom of a glass container and immersed in cyclohexane. A 3 μL droplet of water was deposited onto the surface of the samples and the three-phase contact angle was measured.

## Non-stirred catalytic reduction of 4-nitrophenol using SiO₂NP-metal NP Pickering emulsions

Pickering emulsion catalysts we prepared by shaking 8 mL of aqueous colloid solution with 2.4 mL of dichloromethane and 100 μL of 1 mM $TBA^+NO_3^-$ for 30 s. The aqueous colloid solution was prepared by mixing an appropriate amount of noble metal NP colloid (1 mL for citrate-reduced AgNPs, 2 mL for monodisperse citrate-reduced AuNPs

and 1.5 mL for monodisperse citrate-capped PtNPs) with 0.5 mL of $SiO_2$ colloid, and an appropriate amount of DDI water, so that the total volume of the aqueous solution was 8 mL. After the successful formation of Pickering emulsion, 4 mL of the bulk aqueous phase was removed and replaced with equal volumes of pure DDI water using a pipette. This process was repeated for two times to remove any residual NPs in the aqueous phase. 800 μL of the bulk aqueous phase of the emulsion was removed and replaced with equal volumes of $10^{-3}$ M 4-nitrophenol (aq.). This produced stable o/w emulsions with a slightly yellow colored bulk aqueous phase. To initiate the reaction, 400 μL of $10^{-1}$ M $NaBH_4$ was added to the aqueous phase of the emulsion system. The reaction was carried out at room temperature (ca. 15 °C). The progress of the catalytic reaction was monitored with UV-vis spectroscopy using an Agilent 8453 photodiode array UV–vis spectrophotometer. UV-vis spectroscopy of the initial blank control sample was performed by analyzing 3 mL of $10^{-4}$ M 4-nitrophenol solution. The catalytic reaction was monitored by extracting 3 mL of the aqueous phase from the emulsion system every 5 (10 or 30) min for analysis. After the analysis, the solution was poured back immediately into the reaction system. Optimization experiments were conducted by changing the amounts of $SiO_2$ (0.3 mL and 0.7 mL of $SiO_2$), $NaBH_4$ (300 μL and 500 μL of $NaBH_4$) and reaction temperature (35 °C) using the same protocol as above.

Reduction of 4-nitrophenol attempted with modifier capped AgNP Pickering emulsions as catalysts were performed by adding 100 μL of $10^{-1}$ M $NaBH_4$ (aq.) into Pickering emulsions formed with 3-MPA-capped citrate-reduced AgNPs synthesized via the protocol described above.

## Statistics and reproducibility

All data presented were representative of at least three independent measurements.

## Reporting summary

Further information on research design is available in the Nature Portfolio Reporting Summary linked to this article.

# Data availability

The data that support the findings of this study are available from the corresponding author upon request.

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

## Acknowledgements

X.X. acknowledges National Natural Science Foundation of China (NSFC 21991130; 22233002) for funding support. Y.Z. acknowledges the Chinese Scholarship Council (202008370188) for funding support. W.A. acknowledges the Ministry of Education in Saudi Arabia for funding support. Y.X. acknowledges the Leverhulme Trust Early Career Fellowship (grant ECF2020703) and RSC Researcher Mobility Grant (grant no. RM1602-4142). C.W. acknowledges the European Union's Horizon 2020 research and innovation program under the Marie Skłodowska-Curie grant agreement No 823745 for support. The authors would like to thank Prof Sai Duan and Prof Igor Ying Zhang for their support during this project.

## Author contributions

Y.X., S.B., Z.Y., and Y.Z. contributed to the conceptualization of this work. Y.Z., Z.Y., Y.X., C.L., Q.C., and Y.H. contributed to the development of the methodology. Y.X., S.B., Y.Z. and C.L. contributed to formal analysis. S.B., Y.X., X.X., and C.W. contributed to providing resources. Y.X. and Y.Z. wrote the original draft of the manuscript. Y.X., S.B., Y.Z., Z.Y., C.L., X.X., C.W., and W.A. contributed to reviewing and editing of the draft. Y.X., C.W., and S.B. performed supervision. Y.X. administrated the project. Y.X., S.B. X.X., and C.W. contributed to acquiring the funding for this work.

## Competing interests

The authors declare no competing interests.
