## [Peer Review File · Nature Communications]

General approach to surface-accessible plasmonic Pickering emulsions for SERS sensing and interfacial catalysisReviewer comments, first round review –

Reviewer #1 (Remarks to the Author):

The authors developed a universal modifier-free approach to construct Pickering emulsions carrying fully customizable nano- or microparticle components. The emulsion system was then applied for sensing of some analytes and catalyze 4-nitrophenol (4-NP) to 4-aminophenol (4-AP) reaction. However, the novelty of the study has been reported in other publications JACS, 2021,143,6071; 2019,141(13):5220; 2016;138(32):10173.

In case of catalysis, I recommend the stability testing, as present one is a rather primitive way to assess the colloidal stability of the prepared Pickering emulsions. I would recommend for the authors to use more relevant techniques, such as turbidimetry (Turbiscan), or electrophoretic mobility (zeta potential). Please show the optimization experiments for catalytic reduction (such as reaction conditions, reaction time, catalyst amount, reaction rate, product separation method, etc.) indicating overall catalytic performance of as-prepared pickering emulsion and how the conversion of the reaction was determined? Please provide the H1 NMR spectroscopy data if available. Pickering emulsion system was also applied for the sensing applications however, no calibration curve is found to predict the linear dynamic range and validity of the method. In addition, selectivity and sensitivity should also be addressed. Some typo and language errors are found for examples line number 44, 45, 57-58, 63-64, etc. There are some long and vague sentences which make context unclear.

Reviewer #2 (Remarks to the Author):

In the current work, the authors have established a novel route of preparing pickering emulsions without employing molecular modifiers. The wettability and other surface properties of nanoparticles were shown to be influenced by other solid nanoparticles where the surface of the stabilizer particles (Primary ones) becomes free of modifiers so its reactivity increases. The work is novel, and shows a unique way to prepare pickering emulsions for important applications such as biphasic catalysis with enhanced catalytic activity, further emphasizing the efficiency of molecular modifier free pickering emulsions. Because of this, the current study is very relevant and will be of general interest to the readers of this journal.

I do have few comments which need to be addressed before publication. Therefore, I recommend a minor revision by the authors. I explain my concerns in more detail below and expect the authors to specifically address each of my comments in their response.

Major comments:

1. The study of contact angle was carried out to show the hydrophobicity of modified nanoparticles' surfaces. I suggest to conduct experiments on the modified substrates to determine the surface energies of those surfaces so a better correlation can be drawn between surface properties and nanoparticle chemistry.
2. Zeta potential study needs to be accomplished on the modified particles to find out surface charge and to establish a strong hypothesis behind the interaction of charged and neutral particles and their effect in stabilizing the emulsions.
3. Mechanism of stabilizing the emulsion by forming a co-assembly is not clear. This part needs to be investigated in detail with solid evidences for supporting the mechanism.

Minor comments:

3. Please check the whole document for grammatical errors.
4. Representation on Figure 3 (A) can be redrawn for better clarity of thoughts.

Reviewer #3 (Remarks to the Author):

In this manuscript, the authors report an approach to synthesizing Pickering emulsions, especially those of noble metal nanoparticles including Au, Ag, and Pt, without the use of strongly adsorbed molecular modifiers. Instead, they use carbon nanotubes as particulate stabilizers of the emulsion and use TMA+ as a promoter to shield electrostatic repulsions and cause the interfacial assembly of the metal nanoparticles. As a result, the metal nanoparticles have a clean surface, beneficial for SERS and catalytic applications. This paper is interesting. This assembly manner for the Pickering emulsion is unconventional. Therefore, this paper might become suitable for publication in Nature Communications after the authors address properly the following concerns.

- (1) Evidence of metal nanoparticle assembly on the surface of the Pickering emulsion is missing. After introducing TMA+, it is possible that Au nanoparticles become large assemblies. However, whether these assemblies are at the W/O interface lacks support from experimental data. The authors observed the nanoparticles by SEM. However, the emulsion collapsed during drying. The co-existence of CNTs and Au nanoparticles cannot affirm that they were both at the interface. Did all Au nanoparticles migrate to the W/O interface? Or only partial?
- (2) This synthesis with TMA+ as a promoter is rather dedicated to specific types of nanoparticles, i.e., citrate-capped, or in more general, negatively charged metal nanoparticles, in theory because it relies on the electrostatic force shielding to cause the nanoparticle assembly. Therefore, it cannot be claimed to be "universal". Please comment or revise. Following this understanding, it is difficult to understand why the same mechanism was applied to the synthesis with PVP-capped Ag nanocubes and Au nanostars. The PVP-capped nanoparticles are colloidal because of polymeric steric hindrance, not by electrostatic force. The colloidal dispersity should be difficult to be affected by the ionic strength of the solution. The synthesis or observation should be rationalized in depth.
- (3) SERS sensitivities rely heavily on the construction of hotspots. Did hotspots form in the authors' emulsion? Any evidence? If otherwise, could the authors construct hotspots in the Pickering emulsions by this strategy?
- (4) Catalysis of 4-NP reduction in the Pickering emulsion does not necessarily mean the reaction is at the W/O interface. If the metal nanoparticles are still in the aqueous phase, the reaction does not show a significant difference from the reported results. Again, please rationalize that the metal nanoparticles are at the interface to trigger catalysis there.
- (5) The authors stated that 4-MBA functionalized Au Pickering emulsion may be used as a pH sensor. Please provide data, or preliminary results, to support this claim.

Reviewer #4 (Remarks to the Author):

The manuscript "A universal approach for the construction of surface accessible emulsions to unlock plasmonic sensing and catalytic applications", evaluates a strategy that uses a promoter-assisted co-assembly approach to construct Pickering emulsions carrying fully customizable nano- or microparticle components. This work provides a step change from the current method since it removes the need for chemical modifiers and in turn resolves the long-standing dilemma of choosing between stability and functionality in the synthesis of Pickering emulsions. However, the conclusions obtained in this paper are not sufficiently demonstrated, and the analysis of the experimental results are not detailed enough. This should be supplemented. My comments regarding experiments, obtained results and following discussions are summarized below:

1. The key to the preparation of Pickering emulsions proposed in his paper is the "promoter" molecules and particulate "stabilizers". But these substances have already been studied and used in previous studies (Refer to the work of Zheng (DOI:10.1016/j.jcis.2022.07.085) and Wu (10.1038/ncomms6929)). In addition, the "surface-accessible Pickering emulsions" proposed in this paper are indeed of great interest to the field of SERS and catalysis. However, there is no direct evidence as to whether the target molecule and the target reactant molecule can actually reach the surface of the functional materials.
2. In line 89-91, the first step to generating stable Pickering emulsions is to create favorable

conditions which allow the charged nanoparticles to overcome interparticle electrostatic repulsion, it can be seen from Fig. 2B that CNT plays the role of overcoming interparticle electrostatic repulsion, but it is not explained in detail why CNT can overcome interparticle electrostatic repulsion. Please provide the necessary explanation.

3. In Fig. 2F, the stability of the CNT-Au emulsions over time is considered. It is well known that in substrates used for SERS detection, the stability of the substrate is important for the detection effect. And whether the emulsion formed by CNT-Au has an advantage in terms of stability compared to the common surface-modified emulsion, such as 3-MPA-capped Au emulsions.

4. The role of promoters, such as tetrabutylammonium (TBA⁺) nitrate, is not clear. Not only is there a lack of control experiments for the promoters, but the G and H images in Figure 2 are very blurred. It is not possible to distinguish whether the nanoparticles (functional materials) are formed directly by adsorption onto CNTs or by forming nano islands and thus assembling into Pickering emulsions. Besides, there is also a lack of effective control experiments regarding SERS, and catalytic application-related experiments.

5. In Fig. 2F, the addition of CNTs with concentrations of 0.025 mg/mL and 0.05 mg/mL have about a two-fold difference in the size of the emulsion. Please refine the concentration gradient.

6. Figure 4b compares the detection sensitivity of 3-MPA-capped Au emulsions and Au-CNT emulsions as SERS substrates. The sulfhydryl molecules in the 3-MPA-capped Au emulsions act as modified molecules that replace the citrate barrier analytes on the AuNP surface and thus affect the detection sensitivity, whereas CNT does not act as a barrier molecule to reach the AuNP surface but does not replace the citrate on the AuNP surface, and CNT only stabilizes to some extent. This method still does not solve the problem that AuNP surfactant affects the sensitivity of SERS detection.

7. In addition to this, the paper also contains some obvious writing errors. For example, the phrase "oil-in-water" in line 107 of the text is not consistent with the presentation of the paper.

Response to Reviewers from Authors

We thank the reviewers for their valuable suggestions which have helped us improve our manuscript. Please find a point-by-point response to the questions raised by the reviewers below. Our response to the reviewers' questions is shown in **black bold text**, while the main changes that we have made in our revision are highlighted in **blue bold text**. To show the changes made in this revision some of our comments also include the modified text or figures that have been copied directly from the revised manuscript document. These are shown in **green bold text**.

Reviewer 1

1. The authors developed a universal modifier-free approach to construct Pickering emulsions carrying fully customizable nano- or microparticle components. The emulsion system was then applied for sensing of some analytes and catalyze 4-nitrophenol (4-NP) to 4-aminophenol (4-AP) reaction. However, the novelty of the study has been reported in other publications JACS, 2021,143,6071; 2019,141(13):5220; 2016;138(32):10173.

The papers referenced by the reviewer represent important progress in the general area of Pickering emulsions, and we have now added these to our list of references, in the revised manuscript.

However, we feel that these references address issues that are quite different from our work. More specifically, the first article mentioned by the reviewer (JACS, 2021,143,6071) is focused on probing the effect of different solvents on catalytic reactions in Pickering emulsions; the latter two articles (JACS, 2019,141(13):5220 and JACS, 2016;138(32):10173) discuss the design of Pickering emulsions systems which act as stable microcompartments for flow reactions. In contrast, as the reviewer corrected summarized, our work is focused on developing a method to create modifier-free emulsions, which carry surface-accessible functional particles at the water-oil interface. We showed that this not only allows Pickering emulsions carrying various types of functional components to be readily synthesized, but also greatly enhances the stability and activity of the emulsions in applications, such as SERS and catalysis. Importantly, in all our applications, the functionality of the emulsion comes from the nanoparticles at the interface, which is fundamentally different from emulsion-based microcompartment systems where the applications take place in the dispersed phase of the emulsions.

2. In case of catalysis, I recommend the stability testing, as present one is a rather primitive way to assess the colloidal stability of the prepared Pickering emulsions. I would recommend for the authors to use more relevant techniques, such as turbidimetry (Turbiscan), or electrophoretic mobility (zeta potential).

We thank the reviewer for this suggestion. We have added the zeta potential values of the nanoparticles in this revision. After carefully considering our options, we decided not to perform turbidimetry or electrophoretic mobility measurements to the emulsion samples, since the average size of our emulsion samples is too large and does not disperse homogeneously in solution. This is why we opted to show the evolution of the average diameter of the Pickering emulsions samples over time, since this is the most direct way to characterize and validate emulsion stability (*Adv. Funct. Mater.* 2022, 32, 2110439; *Chem. Commun.* 2018, 54, 10679). We have added additional data showing the average size of CNT-Au Pickering emulsions before and after being stored for one month as Supplementary Fig. 8. To further demonstrate the stability of the Pickering emulsions used as catalysts discussed in this work, we have now added optical microscopy images of the SiO₂-Ag o/w Pickering emulsions samples before and after catalytic reaction. This is presented in Supplementary Fig. 21 in the revision. From the optical images, it can be seen that the average size of the Pickering emulsions is identical before and after reaction, which shows that they remain stable during the reaction process. To showcase the stability of the Pickering emulsions as catalysts, we have also performed new experiments to show that the catalytic activity of the Pickering emulsions is fully retained even when the emulsion samples were stored for a month. This is presented in Fig. 6f and Supplementary Fig. 23 in the revision, which for convenience is also shown below.

Supplementary Figure 8 | Microscopic imaging for characterization of the stability of CNT-Au emulsions. (a)-(b) Optical image and average size of a typical batch of CNT-Au emulsions formed with dichloromethane as the oil phase containing 0.05 mg mL^{-1} of CNT before and after being stored for one month. The scale bars in the optical images correspond to $500 \text{ }\mu\text{m}$.

Supplementary Figure 21 | Probing the stability of $\text{SiO}_2\text{-Ag}$ emulsions after acting as interfacial catalysts. Optical microscopy images of a typical batch of $\text{SiO}_2\text{-Ag}$ emulsion before and after catalytic reduction of 4-NP. The plot shows the average diameter and standard deviation of the emulsion droplets measured from the two images. The scale bars in the optical images correspond to $500 \text{ }\mu\text{m}$.

Figure 6 | Applications of o/w SiO₂-Ag Pickering emulsions in catalysis. (f) Plot showing the signal intensity of 4-NP measured by UV-vis spectroscopy at different points of catalytic reduction using fresh SiO₂-Ag Pickering emulsions and SiO₂-Ag Pickering emulsions that have been stored for 1 month as the catalyst. Inset compares the conversion rate of 4-NP using fresh and stored SiO₂-Ag emulsions.

Supplementary Figure 23 | Comparing the catalytic activity of fresh and stored SiO₂-Ag emulsions. (a) Optical image of a typical batch of freshly prepared SiO₂-Ag emulsions. (b)-(c) UV-vis spectra of 4-NP measured at different stages of the catalytic reduction using freshly prepared SiO₂-Ag emulsions as the catalyst, and the Ln(A/A₀) of 4-NP measured by UV-vis spectroscopy plotted against time. (d) Optical image of a typical batch of SiO₂-Ag emulsions that had been stored at room temperature for 30 days. (e)-(f) UV-vis spectra of 4-NP measured at different stages of the catalytic reduction using SiO₂-Ag emulsions that had been stored for 30 days as the catalyst, and the Ln(A/A₀) of 4-NP measured by UV-vis spectroscopy plotted against time. Panels (b) and (c) are also displayed in Fig. 6 of the main text. Panel (b) is also displayed in Supplementary Fig. 24, 25 and 26.

3. Please show the optimization experiments for catalytic reduction (such as reaction conditions, reaction time, catalyst amount, reaction rate, product separation method, etc.) indicating overall catalytic performance of as-prepared pickering emulsion and how the conversion of the reaction was determined? Please provide the H1 NMR spectroscopy data if available.

The catalytic applications were only meant to demonstrate how removing the modifiers leads to significantly improved functionalities. This was why we did not perform extensive optimization

of the catalytic system in our initial submission, and only showed comparisons between the catalytic activity of the modified and un-modified Pickering emulsions under exactly the same experimental conditions. In this revision we have added experiments to show that the catalytic activity of the surface-accessible Pickering emulsions formed via our new approach can be readily optimized. More specifically, we have performed experiments to probe the stability of the emulsion catalysts and to optimize the reaction temperature, reactant concentration, stabilizer concentration and metal catalyst type. In general, our experimental data was consistent with the findings in literature for the catalytic reduction of 4-nitrophenol. More importantly, our new experimental data demonstrates that even without extensive system optimization, the modifier-free emulsions shown in our work already exhibit excellent functionality. The data for these new experiments are presented in Fig 6, and Supplementary Fig. 21-27 of the revised manuscript. We have also added discussions to demonstrate how the reaction rates and conversion were determined as Supplementary Discussion 2. For convenience the revised figures and discussions are also shown below.

Figure 6 | Applications of o/w SiO₂-Ag Pickering emulsions in catalysis. (a) Schematic illustrations of a modifier-free o/w SiO₂-metal emulsion droplet acting as an interfacial catalyst for the reduction of 4-nitrophenol (4-NP) to 4-aminophenol (4-AP). (b) Optical images of a Ag emulsion sample formed with 3-MPA modifiers before and a few minutes after being treated with NaBH₄ solution. Also shown is the schematic illustrations of the 3-MPA modifiers being replaced by H⁻ in the NaBH₄ solution leading to emulsion destabilization. (c) Optical images of a SiO₂-Ag emulsion sample before and after 20 minutes of catalytic reaction. Also shown is the schematic illustrations of the surface-accessible Ag nanoparticles in modifier-free emulsions acting as stable catalysts. (d) UV-vis spectra of 4-NP measured at different stages of its catalytic reduction using SiO₂-Ag Pickering emulsions as the catalyst. (e) Semi-log plot of Ln(A/A₀) against time for 4-NP. (f) Plot showing the absorbance of 4-NP measured by UV-vis spectroscopy at different points of catalytic reduction using fresh SiO₂-Ag Pickering emulsions and SiO₂-Ag Pickering emulsions that have been stored for 1 month as the catalyst. Inset compares the overall conversion of 4-NP using fresh and stored SiO₂-Ag emulsions.

Supplementary Figure 20 | SiO₂-noble metal emulsions for catalysis. (a)-(c) Optical images of typical batches of SiO₂-Au, Ag and Pt Pickering emulsions, respectively. (d)-(f) SEM images of SiO₂-Au, Ag and Pt Pickering emulsions.

Supplementary Figure 21 | Probing the stability of SiO₂-Ag emulsions after acting as interfacial catalysts. Optical microscopy images of a typical batch of SiO₂-Ag emulsion before and after catalytic reduction of 4-NP. The plot shows the average diameter and standard deviation of the emulsion droplets measured from the two images.

Supplementary Figure 22 | Determination of the reaction order of 4-NP catalytic reduction by SiO₂-Ag emulsions. The signal intensity of 4-NP measured with UV-vis spectroscopy at different points of its catalytic reduction plotted against reaction time.

Supplementary Discussion 2 | Calculation of reaction rate and conversion rate.

The signal intensity of 4-NP is directly proportional to the concentration of 4-NP (Beer-Lambert law). As shown in the data above, the signal intensity of 4-NP plotted versus time can be fitted with an exponential function which shows that it is a pseudo first order rate reaction. Therefore, the reaction rate constant, k , can be calculated following the equation:

$$\ln \frac{A}{A_0} = -kt$$

Plotting $\ln \frac{A}{A_0}$ against t , as shown in Fig. 6e of the main text, gives the value of k_{Ag} , which is 0.1548 min⁻¹.

From Supplementary Fig. S22, it can be seen that the reaction is finished after ca. 15 mins. The overall conversion can be calculated following the equation:

$$\frac{A_{15}}{A_0} \times 100\% = 92\%$$

Supplementary Figure 23 | Comparing the catalytic activity of fresh and stored SiO₂-Ag emulsions. (a) Optical image of a typical batch of freshly prepared SiO₂-Ag emulsions. (b)-(c) UV-vis spectra of 4-NP measured at different stages of the catalytic reduction using freshly

prepared SiO₂-Ag emulsions as the catalyst, and the Ln(A/A₀) of 4-NP measured by UV-vis spectroscopy plotted against time. (d) Optical image of a typical batch of SiO₂-Ag emulsions that had been stored at room temperature for 30 days. (e)-(f) UV-vis spectra of 4-NP measured at different stages of the catalytic reduction using SiO₂-Ag emulsions that had been stored for 30 days as the catalyst, and the Ln(A/A₀) of 4-NP measured by UV-vis spectroscopy plotted against time. Panels (b) and (c) are also displayed in Fig. 6 of the main text. Panel (b) is also displayed in Supplementary Fig. 24-27.

Supplementary Figure 24 | Effect of temperature on the catalytic performance of SiO₂-Ag emulsions. (a)-(d) UV-vis spectra of 4-NP measured at different stages of the catalytic reduction using SiO₂-Ag emulsions as the catalyst at 15 °C and 35 °C, respectively, and the Ln(A/A₀) of 4-NP measured by UV-vis spectroscopy plotted against time. Panels (a) and (b) are also displayed in Fig. 6 of the main text. Panel (a) is also displayed in Supplementary Fig. 23, 25, 26 and 27.

Supplementary Figure 25 | Effect of reactant concentrations on the catalytic performance of SiO₂-Ag emulsions. (a)-(c) UV-vis spectra of 4-NP measured at different stages of the catalytic reduction using SiO₂-Ag Pickering emulsions as the catalyst at various NaBH₄ concentrations. Panel (b) is also displayed in Fig. 6 of the main text and Supplementary Fig. 23, 24, 26 and 27.

Supplementary Figure 26 | Effect of stabilizer concentrations on the catalytic performance of SiO₂-Ag emulsions. (a)-(c) UV-vis spectra of 4-NP measured at different stages of the catalytic reduction using SiO₂-Ag Pickering emulsions as the catalyst at various SiO₂ concentrations. Panel (b) is also displayed in Fig. 6 of the main text and Supplementary Fig. 23, 24, 25 and 27.

Supplementary Figure 27 | Effect of catalyst composition on the catalytic performance of SiO₂-noble metal emulsions. (a)-(c) UV-vis spectra of 4-NP measured at different stages of the catalytic reduction using SiO₂-noble metal Pickering emulsions containing different types of metal nanoparticles. The metal catalysts weight content for Au and Pt were 1.55×10^{-3} and 2.22

$\times 10^{-3}$ (wt.%), respectively. Panel (c) is also displayed in Fig. 6 of the main text and Supplementary Fig. 23, 24, 25 and 26.

We have also modified the main text to reflect the new data mentioned above, which for convenience is also provided below.

“Interfacial catalysis with o/w Pickering emulsions. It is now well-established that strongly adsorbed chemical ligands have a negative impact on the catalytic activity of nanomaterials, since catalytic reactions can only be initiated when the reactants have access to the catalytic surface⁵⁴. This creates a particularly challenging dilemma with traditional emulsion-based nano-catalyst systems, in which emulsion stability and catalytic activity must be carefully balanced. The fact that our current method uncouples the need for the functional particles to provide both stabilization and functionality provides a solution to this long-standing challenge. As shown in Fig. 6a and Supplementary Fig. 20, the catalytic performance of the modifier-free Pickering emulsions was compared with their modifier-capped counterparts using the reduction of 4-nitrophenol (4-NP) with sodium borohydride and o/w SiO₂-noble metal nanoparticle emulsion catalysts as the example.

As shown in Fig. 6b, since the H⁻ ions present in NaBH₄ solutions adsorb strongly to the surface of noble metal nanoparticles and are able to replace even thiols^{54, 55}, the addition of NaBH₄ solution led to the desorption of the thiol modifiers, which immediately destabilized the modifier-capped emulsions. As a result, the emulsions coalesced into several large liquid droplets within minutes. In sharp contrast, since the SiO₂ stabilizers are chemically inert, the SiO₂-metal Pickering emulsions remained completely stable on addition of NaBH₄ solution and throughout the catalytic reaction (Fig. 6c and Supplementary Fig. 21). As shown in Fig. 6d-e, using SiO₂-Ag Pickering emulsions formed with as-prepared citrate-reduced Ag colloid as the functional catalyst, the catalytic reduction of 4-NP with NaBH₄ followed a pseudo first order with a rate constant (k_{Ag}) of ca. $2.58 \times 10^{-3} \text{ s}^{-1}$ and a conversion rate of 92% (see Supplementary Fig. 22 and Supplementary Discussion 2 for details). As discussed in Supplementary Fig. 4 and 10, since a negligible amount of metal nanoparticles remain in the aqueous phase after emulsion formation, the catalytic activity of the emulsions can be attributed solely to the metal nanoparticles at the water-oil interface. Supplementary Table 2 compares the catalytic performance of the unoptimized SiO₂-Ag Pickering emulsions with other state-of-the-art emulsion-based interfacial catalysts designed specifically for the reduction of 4-NP using NaBH₄.⁵⁶⁻⁶⁰ In general, even though the SiO₂-Ag Pickering emulsions were unoptimized and contained ca. 6-19× less metal catalyst, they showed comparable or faster reaction rates compared to the modified emulsions constructed using carefully optimized nanostructures and metal compositions. In addition to the superior catalytic activity, the modifier free emulsions also showed excellent stability over time. As shown in Fig. 6f and Supplementary Fig. 23, the SiO₂-Ag emulsions fully retained their catalytic activity even when left at room temperature for 1 month. Finally, the generality of our co-assembly approach and stability of the product emulsions means that the composition of the emulsions and the reaction conditions can be readily adjusted to suit specific catalytic reactions. For example, Supplementary Fig. 24-27 shows the catalytic performance of the modifier-free emulsions tested at different reaction temperatures, reactant concentrations, stabilizer concentrations and catalyst composition. Consistent with literature, it was found that the rate constant and conversion rate of the reaction was highly dependent on the experimental parameters mentioned above. For example, k_{Ag} increased to ca. $4.24 \times 10^{-3} \text{ s}^{-1}$ when the reaction temperature was simply increased from 15° to 35°, which shows the potential of our approach in the construction of highly active biphasic catalytic systems.”

4. Pickering emulsion system was also applied for the sensing applications however, no calibration curve is found to predict the linear dynamic range and validity of the method. In addition, selectivity and sensitivity should also be addressed.

Similar to the case with the catalysis demonstrations, we did not include extensive optimizations of the SERS studies since the focus was to show that removing the modifiers led to significant improvements in the functionality of the Pickering emulsions. Using adenine and crystal violet as the model analytes, in this revision we have added concentration-dependent SERS data obtained using the modifier-free Pickering emulsions as SERS substrates. In general, the surface-accessible Pickering emulsions showed high plasmonic activities which allowed SERS quantitation to be achieved. As shown in Supplementary Fig. 15, even with simple citrate-reduced Au nanoparticles as the plasmonic component, the limit of detection for adenine and

crystal violet could reach 10^{-7} M and 5×10^{-7} M, respectively. The linear quantitation ranges were found to be 10^{-4} - 10^{-7} M and 10^{-5} - 5×10^{-7} M for adenine and crystal violet, respectively. For convenience Supplementary Fig. 15 is also shown below.

Supplementary Figure 15 | SERS Quantitation with CNT-Au emulsions. (a)-(b) SERS quantitation of crystal violet and adenine using CNT-Au emulsions as the enhancing substrate. Insets show the calibration curves obtained by plotting $\text{Log}(C_{\text{analyte}})$ versus $\text{Log}(I_{\text{analyte}})$.

We have also modified the main text to reflect the new data mentioned above, which for convenience is also provided below:

“In sharp contrast, the surface of the Au nanoparticles in the CNT-Au emulsions only contained weakly adsorbed citrate and chloride capping ligands, which could be easily displaced by adenine (Supplementary Fig. 14). As a result, clear SERS signals of adenine could be obtained even when its concentration was $1000\times$ lower (10^{-7} M) than the concentration used for modifier-capped emulsions (Fig. 5b, spectra *iii-vi*). Additional data on the performance of the CNT-Au emulsions as enhancing substrates for quantitative SERS is shown in Supplementary Fig. 15. Briefly, even with simple citrate-reduced Au nanoparticles as the plasmonic component, the limits of detection for adenine and crystal violet could reach 10^{-7} M and 5×10^{-7} M, respectively. The linear quantitation range was determined to be from 10^{-4} - 10^{-7} M and 10^{-5} - 5×10^{-7} M for adenine and crystal violet, respectively.”

In general, since SERS is a molecularly specific technique, it can be used to simultaneously identify multiple types of analyte molecules that are on the surface of the plasmonic substrate. This was demonstrated in Fig. 5c, spectra *iv*, where the Pickering emulsions were used as biphasic substrates to simultaneously detect adenine and naphthalene. We agree with the reviewer that in some cases, the SERS substrates are functionalized so that it exhibits molecularly specific interactions with a particular type of target molecule. This kind of selectivity can be important for applications, such as bio-analysis where bio-molecules compete with the analyte for enhancing surface. In this regard, the preformed plasmonic hot spots in our emulsion system can act as filters which excluded large biomolecules and selectively allows the adsorption of small molecules for SERS. This effect is currently being studied in our group.

5. Some typo and language errors are found for examples line number 44, 45, 57-58, 63-64, etc. There are some long and vague sentences which make context unclear.

We thank the reviewer for these suggestions. We have now corrected these typos.

Reviewer 2

1. In the current work, the authors have established a novel route of preparing pickering emulsions without employing molecular modifiers. The wettability and other surface properties of nanoparticles were shown to be influenced by other solid nanoparticles where the surface of the stabilizer particles (Primary ones) becomes free of modifiers so its reactivity increases. The work is novel, and shows a unique way to prepare pickering emulsions for important applications such as biphasic catalysis with enhanced catalytic activity, further emphasizing the efficiency of molecular modifier free pickering emulsions. Because of this, the current study is very relevant and will be of general interest to the readers of this journal.

We thank the reviewer for these very favourable comments. We have made significant modifications to our manuscript in this revision following the reviewer's suggestions.

2. The study of contact angle was carried out to show the hydrophobicity of modified nanoparticles' surfaces. I suggest to conduct experiments on the modified substrates to determine the surface energies of those surfaces so a better correlation can be drawn between surface properties and nanoparticle chemistry.

We thank the reviewer for this suggestion. In this revision, we have revised Fig. 1 in the main text and added new experiments (see Supplementary Fig. 1-4, Supplementary Table 1 and Supplementary Discussion 1 in the revised Supplementary Information), along with new discussions to correlate the surface chemistry (wettability) of the particles with the total system energy of our emulsions systems to illustrate the effect of the particle stabilizers and promoters in detail.

In general, the construction of a stable Pickering emulsion system involves, (1) the stabilization of nanoparticles at the water-oil interface and (2) the stabilization of emulsion droplets using the nanoparticle layer. Factor (1) can be considered by analysing the adsorption energy, ΔG , of solid nanoparticles sitting at the interface between water and oil. Here, the main conclusion is that even though it is generally energetically favourable for a solid particle to adsorb to the interface between two immiscible oils, spontaneous interfacial self-assembly is often prohibited by interparticle electrostatic repulsion. Therefore, the key to inducing self-assembly of the charged colloidal nanoparticles demonstrated in this work is to be able to lower the electrostatic repulsion between adjacent particles at the interface so that the process remains energetically favourable. This can be achieved with the addition of promoters which sit between adjacent nanoparticles at the interface to screen electrostatic repulsion. Factor (2) can be considered by analysing the maximum capillary pressure that can be endured by the nanoparticle bilayer, P_c^{max} . A higher P_c^{max} means that the nanoparticle bilayers of two closely placed emulsions droplets are able to withstand a stronger pressing force, which leads to more stable emulsions droplets. Here, the main conclusion is that the value of P_c^{max} increases as the nanoparticles become more hydrophobic for w/o emulsions and more hydrophilic for o/w emulsions. Importantly, this allows the surface-chemistry of the nanoparticles layer to be directly correlated with emulsion stability and accounts for the significant increase in the emulsion stability observed with the addition of "stabilizer" particles.

We have modified both the manuscript and the Supplementary Information in this revision to include the discussion above, which for convenience is also provided below. Since the added discussion is quite long, only the parts directly related to surface-chemistry is included below.

"In general, the construction of a stable Pickering emulsion system involves, (1) the stabilization of nanoparticles at the water-oil interface and (2) the stabilization of emulsion droplets using the nanoparticle layer. First, we analyse (1) by considering the adsorption energy, ΔG , of a spherical solid nanoparticle sitting at the interface between water and oil²³:

$$\Delta G = -\pi R^2 \gamma_{wo} (1 \pm \cos \theta)^2 \quad (1)$$

where R is the radius of the nanoparticle, γ_{wo} is the surface tension between the oil and water, θ is the three-phase contact angle measured through the dispersed phase on a solid particle in the environment of the dispersion medium and "+" and "-" signs refer to w/o emulsions and o/w emulsions, respectively.

Equation (1) shows that the adsorption of nanoparticles to the water-oil interface is generally a favourable process, provided that θ is not 0° or 180° . Indeed, spontaneous interfacial self-assembly was observed for the uncharged CNTs dispersed in oil, when shaken with water. However, for electrostatically stabilized colloidal nanoparticles, the overall change in system energy due to self-assembly can no longer be described using just Equation (1), which is based solely on the reduction in interfacial energy. The additional gain in electrostatic potential from the nanoparticles packing closely at the interface must be considered, which can be expressed as below²⁶.

.....

In addition to the successful assembly of nanoparticles at the water-oil interface, the construction of stable Pickering emulsions also requires condition (2) to be met, which is that the particle layers at the interface need to be able to stabilize the thin liquid layer that is formed when two emulsion droplets move close to each other. This condition can be considered as a requirement for the nanoparticle bilayers to be able to withstand the pressing force between two approaching emulsion droplets. For densely packed solid spherical nanoparticles, the maximum capillary pressure that can be withstood by the nanoparticle bilayer, P_c^{max} can be defined as³¹:

$$P_c^{max} = \pm p \frac{2\gamma_{wo}}{R} (\cos \theta \pm z) \quad (4)$$

with the emulsion system becoming more stable at higher P_c^{max} values. In the equation, the sign “+” refers to o/w emulsions, and sign “-” refers to w/o emulsions, while p and z are parameters, which have positive values that increase with higher particle coverage at the interface (Supplementary Table S1).

Importantly, Equation (4) shows that the value of P_c^{max} increases as the nanoparticles become more hydrophobic for w/o emulsions and more hydrophilic for o/w emulsions. As shown in Fig. 1d, θ_{Au} and θ_{CNT} were measured to be 68° and 113° , respectively. Although the wettability of both Au nanoparticles and CNTs satisfy the minimum θ_{min} threshold of 50.7° required to stabilize w/o emulsions³¹, the maximum pressing force that two CNT stabilized w/o emulsions could withstand was calculated to be ca. 5× higher than for the Au stabilized w/o emulsions (Supplementary Discussion 1). As a result, stable Pickering emulsions could only be obtained using CNTs, while emulsions formed with citrate-stabilized Au nanoparticles coalesced in seconds into an interfacial film, as shown in Fig. 1e and f. Mixing the CNTs with Au nanoparticles led to the formation of a mixed interfacial nanoparticle layer whose hydrophobicity resembled that of the plain CNTs rather than the Au nanoparticles ($\theta_{CNT-Au} \approx \theta_{CNT} > \theta_{Au}$). This allowed the CNTs to act as emulsion stabilizers for constructing highly stable w/o Pickering emulsions that contained a layer of CNTs and Au nanoparticles at the interface, as shown in Fig. 1g and i. It is worth noting that, in practice, the increase in the P_c^{max} value with the addition of CNTs will be even higher than the value estimated above, since it has been shown that CNTs are able to cover a significantly higher proportion of surface area compared to the spherical geometry assumed for all particles in Equation (4), which would lead to increased p and z values³².

3. Zeta potential study needs to be accomplished on the modified particles to find out surface charge and to establish a strong hypothesis behind the interaction of charged and neutral particles and their effect in stabilizing the emulsions.

The zeta potential of the charged nanoparticle colloids, such as SiO₂, citrate-stabilized Au, Pt and Ag are at least -40 mV. We have added this information to the revised manuscript. As discussed in the Question above, we believe that the particle “stabilizers” stabilize the emulsions by changing the overall wettability of the particle layer. In essence, this means they stabilize the emulsions independently of the “functional nanoparticles”, and without completely occupying the interfacial area, which allows the unmodified functional particles to be filled onto the surface of the stabilized emulsion. Therefore, we do not believe that the “particle stabilizers” interact with the “functional particles” and alter their surface charge. Taking CNT-Au emulsions as an example, the CNT and Au nanoparticles are dispersed in the oil and water phase, respectively (as shown in Fig. 1a of the revised manuscript). Upon adding the promoter and shaking, the CNTs and Au nanoparticles migrate to the water-oil interface independently of each other. We note that without the promoters, the uncharged CNTs are able to assemble at the interface since this is energetically favourable but the charged Au nanoparticles cannot since

the gain in electrostatic potential that would arise from packing the charged Au nanoparticles densely at the interface makes self-assembly energetically unfavourable. This effect is generally observed for all the particles used in this work i.e. only uncharged particles can migrate spontaneously to the interface without promoters.

To explain the above in detail, we have modified Fig. 1 and added corresponding discussion in the main text, which for convenience is shown below.

Figure 1 | The working principles of “stabilizers” and “promoters”. (a) Schematic illustration of the experimental process for generating carbon nanotube (CNT)-Au emulsions. (b)-(c) Schematic illustrations of the working principle of promoters. (d) Three-phase contact angle of CNTs, Au and CNT-Au nanoparticle layers measured for water-cyclohexane. (e)-(g) Photographs of CNT emulsions, Au films, and CNT-Au emulsions. (h)-(i) Schematic illustrations of the working principle of CNT stabilizers.

“Equation (1) shows that the adsorption of nanoparticles to the water-oil interface is generally a favourable process, provided that θ is not 0° or 180° . Indeed, spontaneous interfacial self-assembly was observed for the uncharged CNTs dispersed in oil, when shaken with water. However, for electrostatically stabilized colloidal nanoparticles, the overall change in system energy due to self-assembly can no longer be described using just Equation (1), which is based solely on the reduction in interfacial energy. The additional gain in electrostatic potential from the nanoparticles packing closely at the interface must be considered, which can be expressed as below²⁶.

$$\mu_{coulomb,w} = \frac{z\pi R^3 \sigma^2}{8\epsilon_w \epsilon_0} \left(3\sqrt{y} + \frac{2R}{L_D} \right) e^{L_D \sqrt{y}} \quad (2)$$

$$\mu_{coulomb,o} = \frac{3z\pi R^3 \alpha^2 \sigma^2 \sqrt{y}}{8\epsilon_{oil} \epsilon_0} \quad (3)$$

More specifically, Equations (2)-(3) show the electrostatic potential between two similarly charged spherical nanoparticles within a hexagonally packed nanoparticle-lattice on either side of the interface (Supplementary Fig. 1), where $\mu_{coulomb,w}$ and $\mu_{coulomb,o}$ are the electrostatic potential between the parts of the nanoparticles immersed in the water and oil phases, respectively, z is the number of nearest neighbouring particles, y is the fraction of interface covered by nanoparticles versus the maximum fraction of interface that could be covered by the nanoparticles, σ is the particle charge density, ϵ_{oil} and ϵ_w are the dielectric constants of the oil and water phase, and ϵ_0 is the permittivity of vacuum. Since the charged species on the surface of the nanoparticles are inherently hydrophilic, it is unlikely that they can be completely transferred across the interface into the oil phase. Additionally, some of the charged species which remain on the surface of the particles could recombine with their counter-ion upon

entering the oil phase. Therefore, a scaling factor α , which ranges between 0 and 1, is introduced in Equation 3.

In practice, this means that electrostatically stabilized colloidal nanoparticles, such as the Au nanoparticles used here, do not migrate spontaneously to the interface when the colloid is simply shaken with an immiscible oil (Fig. 1b and Supplementary Fig. 2). Therefore, the key to inducing self-assembly of charged colloidal nanoparticles at water-oil interfaces is to lower the electrostatic repulsion between adjacent particles at both sides of the interface. As discussed above, the conventional approach to achieve this is through the addition of organic modifiers, such as thiols, or co-solvents, such as ethanol, but modifiers passivate the surface of the functional nanoparticles while co-solvents destabilize emulsion systems. Previously, we and others have shown that an alternative approach to induce self-assembly of charged colloidal nanoparticles is to use promoters, such as tetrabutylammonium (TBA⁺) nitrate, which are lipophilic or amphiphilic organo-electrolytes that carry an opposite charge to the nanoparticles²⁷⁻²⁹. As shown in Fig. 1c, the promoters along with the counterions, such as Na⁺, which are initially present in the colloidal solution sit between the charged nanoparticles at each side of the interface to provide charge-screening which helps overcome interparticle electrostatic repulsion. It is useful to note that this method is also fundamentally different from previous reports which use simple hydrophilic salts, such as NaCl, to induce partial aggregation of the colloid and therefore lower interparticle electrostatic repulsion at the interface³⁰. Here, the addition of promoters does not lead to any observable aggregation as shown by UV-vis spectroscopy (Supplementary Fig. 3). Therefore, using promoters allows the charged colloidal nanoparticles to be assembled into densely packed superstructures without passivating their surface with strongly adsorbed modifiers or inducing undesired aggregation. In principle, the minimum amount of promoter required to effectively provide charge screening will differ depending on the surface-charge of the colloid used for self-assembly. However, this is not an issue in practice since a reasonable excess of promoter can be used routinely without perturbing the emulsion system. We have found that a promoter concentration of ca. 3×10^{-5} M (per volume of colloid) was adequate for inducing self-assembly of highly charged colloids with zeta potential ca. -40 mV²⁸ (Supplementary Fig. 4). Based on the above, a promoter concentration of 2.4×10^{-3} M was consistently used for all emulsion systems in this work to mitigate interparticle electrostatic repulsion during self-assembly.”

In addition, we have also added new experimental data in the Supplementary Information, which for convenience is shown below.

[Redacted]

Supplementary Figure 1 | Schematic illustration of a densely packed hexagonal array of spherical nanoparticles. R corresponds to the radius of the particles, x corresponds to the surface-to-surface distance between two adjacent particles. Figure reproduced from Ref 8.

Supplementary Figure 2 | Demonstration that promoters are essential for self-assembly. Optical images showing a sample of citrate-reduced Ag nanoparticles and SiO₂ nanoparticles before and after being shaken with dichloromethane without any promoters. The sample remains unchanged, and the nanoparticles do not migrate to the water-oil interface. SiO₂ stabilizers were selected as an example since they are charged and dispersed in the aqueous phase.

Supplementary Figure 3 | Probing the influence of tetrabutylammonium (TBA) salt on colloid stability. UV-vis spectra of citrate-reduced Ag and Au colloid with and without the addition of TBA promoters. The concentration of promoters was the same as that used for the synthesis of Pickering emulsions.

Supplementary Figure 4 | Demonstration of the migration of nanoparticles to the water-oil interface in promoter-assisted self-assembly. (a)-(b) Optical images showing a sample of citrate-reduced Ag nanoparticles and SiO₂ nanoparticles before and after being shaken with dichloromethane and promoters. (c) UV-vis spectra of the aqueous colloid phase before and after self-assembly showing that a negligible number of nanoparticles remain in the bulk aqueous phase after the formation of Pickering emulsions.

- Mechanism of stabilizing the emulsion by forming a co-assembly is not clear. This part needs to be investigated in detail with solid evidences for supporting the mechanism.

Following the reviewers' advice, we have now updated the main text to include a comprehensive analysis of our Pickering emulsion system along with new experimental data in this revision. More specifically, we analysed the role of the "particle stabilizers" and "promoters" in regards to the adsorption energy, electrostatic repulsion, and maximum capillary pressure of the emulsion system. This allowed us to establish a more detailed relationship between the surface chemistry of the nanoparticle layer and the stability of the emulsions and to clearly identify the roles of promoters and stabilizers in the self-assembly process. The modifications made to the manuscript and Supplementary Information are presented and discussed in detail above, in the responses to Questions 2 and 3 by Reviewer 2. In addition, to better reflect the role of "particle stabilizers" and "emulsions", we have also modified Fig. 1, in the revised main text, which is also shown in the previous comment.

- Please check the whole document for grammatical errors.

We have checked our documents for typos and errors.

- Representation on Figure 3 (A) can be redrawn for better clarity of thoughts.

We have revised Figure 3(a) (now Fig. 4a in the revision) to improve the clarity of the figure. For convenience, the revised Figure is provided below.

Figure 4 | Expanding Pickering emulsion types via promoter assisted co-assembly. (a) Schematic illustrations showing the key interchangeable components within the promoter assisted co-assembly of Pickering emulsions.

Reviewer 3

1. In this manuscript, the authors report an approach to synthesizing Pickering emulsions, especially those of noble metal nanoparticles including Au, Ag, and Pt, without the use of strongly adsorbed molecular modifiers. Instead, they use carbon nanotubes as particulate stabilizers of the emulsion and use TMA+ as a promoter to shield electrostatic repulsions and cause the interfacial assembly of the metal nanoparticles. As a result, the metal nanoparticles have a clean surface, beneficial for SERS and catalytic applications. This paper is interesting. This assembly manner for the Pickering emulsion is unconventional. Therefore, this paper might become suitable for publication in Nature Communications after the authors address properly the following concerns.

We thank the reviewer for these highly favourable comments. We have made significant modifications to our manuscript in this revision following the reviewer's suggestions.

2. Evidence of metal nanoparticle assembly on the surface of the Pickering emulsion is missing. After introducing TMA+, it is possible that Au nanoparticles become large assemblies. However, whether these assemblies are at the W/O interface lacks support from experimental data. The authors observed the nanoparticles by SEM. However, the emulsion collapsed during drying. The co-existence of CNTs and Au nanoparticles cannot affirm that they were both at the interface. Did all Au nanoparticles migrate to the W/O interface? Or only partial?

We agree with the reviewer that the migration of the nanoparticles to the interface is a key aspect to our work. In general, we believe the particles are at the interface since this would be essential for the formation of stable Pickering emulsions and for the biphasic SERS applications demonstrated in our work. To systematically demonstrate this, we have added new experimental data in the revised Supplementary Information as Supplementary Fig. 2-4 and modified the main text accordingly in this revision. More specifically, we showed that without the addition of promoters, the charged colloidal nanoparticles remain in the bulk aqueous phase after shaking, and as a result, no emulsions are formed. Conversely, with the addition of promoters, the aqueous phase becomes clear and colourless after the formation of emulsions, which shows that the majority of metal nanoparticle have migrated to the interface. This is also backed up by UV-vis measurements of the bulk aqueous phase before and after self-assembly, which showed that the amount of metal nanoparticles left in solution is negligible. The above was demonstrated using o/w SiO₂-Au emulsions since this allowed the aqueous phase to be easily observed and extracted for ex-situ analysis, however we have verified that the general conclusions holds true for the various types of emulsions mentioned in this work. For convenience Supplementary Fig. 2-4 are also shown below.

Supplementary Figure 2 | Demonstration that promoters are essential for self-assembly. Optical images showing a sample of citrate-reduced Ag nanoparticles and SiO₂ nanoparticles before and after being shaken with dichloromethane without any promoters. The sample remains unchanged, and the nanoparticles do not migrate to the water-oil interface. SiO₂ stabilizers were selected as an example since they are charged and dispersed in the aqueous phase.

Supplementary Figure 3 | Probing the influence of tetrabutylammonium (TBA) salt on colloid stability. UV-vis spectra of citrate-reduced Ag and Au colloid with and without the addition of TBA promoters. The concentration of promoters was the same as that used for the synthesis of Pickering emulsions.

Supplementary Figure 4 | Demonstration of the migration of nanoparticles to the water-oil interface in promoter-assisted self-assembly. (a)-(b) Optical images showing a sample of citrate-reduced Ag nanoparticles and SiO₂ nanoparticles before and after being shaken with dichloromethane and promoters. (c) UV-vis spectra of the aqueous colloid phase before and after self-assembly showing that a negligible number of nanoparticles remain in the bulk aqueous phase after the formation of Pickering emulsions.

3. This synthesis with TMA⁺ as a promoter is rather dedicated to specific types of nanoparticles, i.e., citrate-capped, or in more general, negatively charged metal nanoparticles, in theory because it relies on the electrostatic force shielding to cause the nanoparticle assembly. Therefore, it cannot be claimed to be "universal". Please comment or revise.

We agree with the reviewer that that TBA⁺, or more generally, positively charged promoters are only useful for inducing self-assembly of negatively charged colloidal nanoparticles. Therefore, we have changed “universal” to “general” in this revision. We believe our method is very general, since we showed that our approach is effective for constructing surface-accessible Pickering emulsions from both charge neutral (also mentioned in the next question by the reviewer) and negatively charged nanoparticles. This already accounts for the majority of nanoparticles, since there are far fewer types of nanoparticles which inherently carry a positive surface charge. For the sake of the current discussion, we believe that the concept of using “promoters” to induce

particle assembly at the water-oil interface is, in theory, universal, since positively charged nanoparticles can be coupled with negatively charged promoters, such as, tetraphenylborate (TPB-) in self-assembly. This was demonstrated in our previous work in interfacial self-assembly of nanoparticle films, using colloidal nanoparticles which were deliberately modified to have a positive surface charge (*Nano Lett.* 2016, 16, 5255).

4. Following this understanding, it is difficult to understand why the same mechanism was applied to the synthesis with PVP-capped Ag nanocubes and Au nanostars. The PVP-capped nanoparticles are colloidal because of polymeric steric hindrance, not by electrostatic force. The colloidal dispersity should be difficult to be affected by the ionic strength of the solution. The synthesis or observation should be rationalized in depth.

The reviewer is right that for common charge-neutral nanoparticles, such as PVP capped Au nanostars and Ag nanocubes, only the particle “stabilizers” are required for the formation of stable Pickering emulsions. We did not specify this since the promoters do not introduce any adverse effects to the emulsion system and we wanted to present a unified modifier-free approach that can be easily applied even by non-experts. However, we realize that without an appropriate discussion, this is inaccurate and can be confusing. Therefore, we have now made significant modifications to the main text and Supplementary Information in this revision to include the discussions made here and a detail analysis of the mechanism of the promoters and stabilizers. For convenience the most important changes to the discussions in the main text are presented below.

“Equation (1) shows that the adsorption of nanoparticles to the water-oil interface is generally a favourable process, provided that θ is not 0° or 180° . Indeed, spontaneous interfacial self-assembly was observed for the uncharged CNTs dispersed in oil, when shaken with water. However, for electrostatically stabilized colloidal nanoparticles, the overall change in system energy due to self-assembly can no longer be described using just Equation (1), which is based solely on the reduction in interfacial energy. The additional gain in electrostatic potential from the nanoparticles packing closely at the interface must be considered, which can be expressed as below²⁶.

$$\mu_{coulomb,w} = \frac{z\pi R^3 \sigma^2}{8\varepsilon_w \varepsilon_0} \left(3\sqrt{y} + \frac{2R}{L_D} \right) \frac{2R}{L_D} e^{L_D \sqrt{y}} \quad (2)$$

$$\mu_{coulomb,o} = \frac{3z\pi R^3 \alpha^2 \sigma^2 \sqrt{y}}{8\varepsilon_{oil} \varepsilon_0} \quad (3)$$

More specifically, Equations (2)-(3) show the electrostatic potential between two similarly charged spherical nanoparticles within a hexagonally packed nanoparticle-lattice on either side of the interface (Supplementary Fig. 1), where $\mu_{coulomb,w}$ and $\mu_{coulomb,o}$ are the electrostatic potential between the parts of the nanoparticles immersed in the water and oil phases, respectively, z is the number of nearest neighbouring particles, y is the fraction of interface covered by nanoparticles versus the maximum fraction of interface that could be covered by the nanoparticles, σ is the particle charge density, ε_{oil} and ε_w are the dielectric constants of the oil and water phase, and ε_0 is the permittivity of vacuum. Since the charged species on the surface of the nanoparticles are inherently hydrophilic, it is unlikely that they can be completely transferred across the interface into the oil phase. Additionally, some of the charged species which remain on the surface of the particles could recombine with their counter-ion upon entering the oil phase. Therefore, a scaling factor α , which ranges between 0 and 1, is introduced in Equation 3.

In practice, this means that electrostatically stabilized colloidal nanoparticles, such as the Au nanoparticles used here, do not migrate spontaneously to the interface when the colloid is simply shaken with an immiscible oil (Fig. 1b and Supplementary Fig. 2). Therefore, the key to inducing self-assembly of charged colloidal nanoparticles at water-oil interfaces is to lower the electrostatic repulsion between adjacent particles at both sides of the interface. As discussed above, the conventional approach to achieve this is through the addition of organic modifiers, such as thiols, or co-solvents, such as ethanol, but modifiers passivate the surface of the functional nanoparticles while co-solvents destabilize emulsion systems. Previously, we and others have shown that an alternative approach to induce self-assembly of charged colloidal nanoparticles is to use promoters, such as tetrabutylammonium (TBA⁺) nitrate, which are

lipophilic or amphiphilic organo-electrolytes that carry an opposite charge to the nanoparticles²⁷⁻²⁹. As shown in Fig. 1c, the promoters along with the counterions, such as Na⁺, which are initially present in the colloidal solution sit between the charged nanoparticles at each side of the interface to provide charge-screening which helps overcome interparticle electrostatic repulsion. It is useful to note that this method is also fundamentally different from previous reports which use simple hydrophilic salts, such as NaCl, to induce partial aggregation of the colloid and therefore lower interparticle electrostatic repulsion at the interface³⁰. Here, the addition of promoters does not lead to any observable aggregation as shown by UV-vis spectroscopy (Supplementary Fig. 3). Therefore, using promoters allows the charged colloidal nanoparticles to be assembled into densely packed superstructures without passivating their surface with strongly adsorbed modifiers or inducing undesired aggregation. In principle, the minimum amount of promoter required to effectively provide charge screening will differ depending on the surface-charge of the colloid used for self-assembly. However, this is not an issue in practice since a reasonable excess of promoter can be used routinely without perturbing the emulsion system. We have found that a promoter concentration of ca. 3×10^{-5} M (per volume of colloid) was adequate for inducing self-assembly of highly charged colloids with zeta potential ca. -40 mV²⁸ (Supplementary Fig. 4). Based on the above, a promoter concentration of 2.4×10^{-3} M was consistently used for all emulsion systems in this work to mitigate interparticle electrostatic repulsion during self-assembly.”

5. SERS sensitivities rely heavily on the construction of hotspots. Did hotspots form in the authors' emulsion? Any evidence? If otherwise, could the authors construct hotspots in the Pickering emulsions by this strategy?

We agree with the reviewer that the formation of plasmonic hot spots is crucial for SERS analysis. In addition to the SEM data and discussion provided in the previous version of the manuscript, we have now performed new TEM and in-situ SERS studies to systematically demonstrate the successful formation of plasmonic hot spots in our emulsion system in this revision. The data is shown in Fig. 3h of the main text and Supplementary Fig. 10 in this revision and is also shown below.

Figure 3 | Microscopic characterization of CNT-Au emulsions. (h) Transmission microscopy image of a typical area in a CNT-Au nanoparticle layer formed using 0.05 mg mL^{-1} of CNTs.

Supplementary Figure 10 | Confocal SERS microscopy showing the localization of nanoparticles and the formation of plasmonic hot spots at the water-oil interface. SERS spectra obtained from the surface (water-oil interface) of a CNT-Au emulsion droplet (i), from the parent Au nanoparticle colloid (ii) and the bulk aqueous phase of the same CNT-Au emulsion droplet (iii).

As shown in Supplementary Fig. 10, the citrate-reduced Au nanoparticles used for the SERS studies in our work only show minimum SERS activity when dispersed in solution as a colloid due to the lack of plasmonic hot spots. However, their SERS signals increase dramatically when the particles have been brought to the interface which suggests the formation of interparticle plasmonic hot spots. Moreover, it can be seen that no observable SERS signals can be obtained when the analysis plane of the confocal microscope is focused on a point in the bulk aqueous solution. This shows that the SERS signals observed from the emulsions arise from SERS active plasmonic hot spots formed by the particles assembled at the interface only.

We have also modified the main text to incorporate the discussions above in this revision, which for convenience is shown below.

“The arrangement of the binary nanoparticle layer was further studied using transmission electron microscopy (TEM) and scanning electron microscopy-energy dispersive X-ray spectroscopy (SEM-EDX) (see Methods in Supplementary for details). Fig. 3h show the TEM image of a typical area of the CNT-Au nanoparticle layer formed using 0.05 mg mL^{-1} of CNTs. In general, it was found that regardless of the composition of the binary nanoparticle layer, the CNTs were always spread out evenly as an entangled mesh while the Au nanoparticles occupied the gaps in the form of islands, which grew in size with increased concentrations of Au in the Pickering emulsions (Supplementary Fig. 9). At the highest relative concentration of Au to CNTs (corresponding to 0.025 mg mL^{-1} of CNTs), the Au nanoparticle islands fused to form larger networks, at which point the stability of the corresponding Pickering emulsions decreased from >30 days to ca. 7 days (Fig. 3g). The arrangement of the particles at the interface was also probed using in-situ confocal SERS spectroscopy. As shown in Supplementary Fig. 10, a dramatic increase in the SERS signal intensity of the capping ligands adsorbed on the Au nanoparticles was observed when the colloidal Au nanoparticles were assembled from solution onto the surface of the emulsions. This suggested the formation of interparticle plasmonic hot spots during self-assembly⁴¹, and is consistent with the ex-situ electron microscopy data shown above. The Au nanoparticles preferred to form islands rather than mix evenly with the CNTs was likely due to the large differences in the surface chemistry of the two types of particles, and has been previously observed in other binary nanoparticle interfacial assemblies⁴². This particle arrangement is highly favourable for generating strong plasmonic properties, which underpin important applications in sensing, catalysis and photodynamic therapy⁴³⁻⁴⁵.”

6. Catalysis of 4-NP reduction in the Pickering emulsion does not necessarily mean the reaction is at the W/O interface. If the metal nanoparticles are still in the aqueous phase, the reaction does not show a significant difference from the reported results. Again, please rationalize that the metal nanoparticles are at the interface to trigger catalysis there.

The reviewer makes an excellent point. We thank the reviewer for this suggestion. In this revision, we have examined this possibility rigorously. As a result, we have improved and updated our protocol for fabricating emulsion catalysts, which is reflected in the Methods section of the revised Supplementary Information document. Correspondingly, we have reperformed all the experiments for the catalytic reduction of 4-NP and updated all the data presented in our manuscript involving catalysis in this revision (see Fig. 6 in the revised main text and Supplementary Fig. 21-27 in the revised Supplementary Information). In addition, we have added new experimental studies to confirm the localization of the functional nanoparticles at the interface and to confirm that the remaining number of nanoparticles in the bulk aqueous phase is negligible (see Supplementary Fig. 4 and 10 in the revised Supplementary Information). We have updated the corresponding section of the main text accordingly to clearly state that the catalytic effect observed in this system arises purely from the nanoparticle catalysts residing at the water-oil interface. For convenience, the updated Fig. 6 and the corresponding parts of the main text are presented below. Supplementary Fig. 4 and 6 and be found in the replies to Questions 2 and 5 by Reviewer 3.

Figure 6 | Applications of o/w SiO₂-Ag Pickering emulsions in catalysis. (a) Schematic illustrations of a modifier-free o/w SiO₂-metal emulsion droplet acting as an interfacial catalyst for the reduction of 4-nitrophenol (4-NP) to 4-aminophenol (4-AP). (b) Optical images of a Ag emulsion sample formed with 3-MPA modifiers before and a few minutes after being treated with NaBH₄ solution. Also shown is the schematic illustrations of the 3-MPA modifiers being replaced by H⁺ in the NaBH₄ solution leading to emulsion destabilization. (c) Optical images of a SiO₂-Ag emulsion sample before and after 20 minutes of catalytic reaction. Also shown is the schematic illustrations of the surface-accessible Ag nanoparticles in modifier-free emulsions acting as stable catalysts. (d) UV-vis spectra of 4-NP measured at different stages of its catalytic reduction using SiO₂-Ag Pickering emulsions as the catalyst. (e) Semi-log plot of Ln(A/A₀) against time for 4-NP. (f) Plot showing the absorbance of 4-NP measured by UV-vis spectroscopy at different points of catalytic reduction using fresh SiO₂-Ag Pickering emulsions and SiO₂-Ag Pickering emulsions that have been stored for 1 month as the catalyst. Inset compares the overall conversion of 4-NP using fresh and stored SiO₂-Ag emulsions.

“As shown in Fig. 6b, since the H⁺ ions present in NaBH₄ solutions adsorb strongly to the surface of noble metal nanoparticles and are able to replace even thiols^{54, 55}, the addition of NaBH₄ solution led to the desorption of the thiol modifiers, which immediately destabilized the modifier-capped emulsions. As a result, the emulsions coalesced into several large liquid droplets within minutes. In sharp contrast, since the SiO₂ stabilizers are chemically inert, the SiO₂-metal Pickering emulsions remained completely stable on addition of NaBH₄ solution and throughout the catalytic reaction (Fig. 6c and Supplementary Fig. 21). As shown in Fig. 6d-e, using SiO₂-Ag Pickering emulsions formed with as-prepared citrate-reduced Ag colloid as the functional catalyst, the catalytic reduction of 4-NP with NaBH₄ followed a pseudo first order with a rate constant (k_{Ag}) of ca. $2.58 \times 10^{-3} \text{ s}^{-1}$ and a conversion rate of 92% (see Supplementary Fig. 22 and Supplementary Discussion 2 for details). As discussed in Supplementary Fig. 4 and 10, since a negligible amount of metal nanoparticles remain in the aqueous phase after emulsion formation, the catalytic activity of the emulsions can be attributed solely to the metal nanoparticles at the water-oil interface. Supplementary Table 2 compares the catalytic performance of the unoptimized SiO₂-Ag Pickering emulsions with other state-of-the-art emulsion-based interfacial catalysts designed specifically for the reduction of 4-NP using NaBH₄.⁵⁶⁻⁶⁰ In general, even though the SiO₂-Ag Pickering emulsions were unoptimized and contained ca. 6-19× less metal catalyst, they showed comparable or faster reaction rates compared to the modified emulsions constructed using carefully optimized nanostructures and metal compositions. In addition to

the superior catalytic activity, the modifier free emulsions also showed excellent stability over time. As shown in Fig. 6f and Supplementary Fig. 23, the SiO₂-Ag emulsions fully retained their catalytic activity even when left at room temperature for 1 month. Finally, the generality of our co-assembly approach and stability of the product emulsions means that the composition of the emulsions and the reaction conditions can be readily adjusted to suit specific catalytic reactions. For example, Supplementary Fig. 24-27 shows the catalytic performance of the modifier-free emulsions tested at different reaction temperatures, reactant concentrations, stabilizer concentrations and catalyst composition. Consistent with literature, it was found that the rate constant and conversion rate of the reaction was highly dependent on the experimental parameters mentioned above. For example, k_{Ag} increased to ca. $4.24 \times 10^{-3} \text{ s}^{-1}$ when the reaction temperature was simply increased from 15° to 35°, which shows the potential of our approach in the construction of highly active biphasic catalytic systems.”

7. The authors stated that 4-MBA functionalized Au Pickering emulsion may be used as a pH sensor. Please provide data, or preliminary results, to support this claim.

The preliminary SERS data showing the potential of 4-MBA functionalized Au Pickering emulsions as pH sensors can be found in Figure S11 of the previous version of the Supplementary Information. We have modified this figure to improve its clarity and it is now shown as Supplementary Fig. 19 in this revision. For convenience the Figure is also presented below.

Supplementary Figure 19 | Demonstrating the potential of surface-functionalized CNT-Au emulsions as SERS pH sensors. SERS of 4-mercaptobenzoic acid (MBA) at different pH values obtained using CNT-Au Pickering emulsions. The SERS intensity of the spectra have been normalized for illustration purposes.

We have modified the main text corresponding to this section to discuss this with more detail and to clearly refer the readers to this Figure. For convenience, the modified text is presented below.

“The fact that the plasmonic nanoparticles are no longer relied upon for providing emulsion stability also means that their surface can be tailored to introduce advanced functionalities without affecting the stability of the emulsions. To illustrate this, the surface of the Au nanoparticles was functionalized with a self-assembled monolayer of 4-mercaptobenzoic acid (4-MBA). Similar to the citrate-capped Au nanoparticles, 4-MBA-capped Au nanoparticles cannot stabilize Pickering emulsions alone but could form highly stable emulsions when co-assembled with CNT stabilizers. As shown in Supplementary Fig. 19, the SERS spectra of 4-MBA showed distinct vibrational bands at different pH values depending on the protonation state of their carboxylic functional groups, which gives the 4-MBA-functionalized plasmonic emulsions the potential to act as pH sensors⁵³.”

Reviewer 4

1. The manuscript “A universal approach for the construction of surface accessible emulsions to unlock plasmonic sensing and catalytic applications”, evaluates a strategy that uses a promoter-assisted co-assembly approach to construct Pickering emulsions carrying fully customizable nano- or microparticle components. This work provides a step change from the current method since it removes the need for chemical modifiers and in turn resolves the long-standing dilemma of choosing between stability and functionality in the synthesis of Pickering emulsions.
However, the conclusions obtained in this paper are not sufficiently demonstrated, and the analysis of the experimental results are not detailed enough. This should be supplemented. My comments regarding experiments, obtained results and following discussions are summarized below.

We thank the reviewer for the highly favourable remarks on the novelty and significance of our work. We have made significant changes in this revision by adding new experimental data and theoretical analysis to further support our conclusions.

2. The key to the preparation of Pickering emulsions proposed in his paper is the "promoter" molecules and particulate "stabilizers". But these substances have already been studied and used in previous studies (Refer to the work of Zheng (DOI:10.1016/j.jcis.2022.07.085) and Wu (10.1038/ncomms6929)). In addition, the "surface-accessible Pickering emulsions" proposed in this paper are indeed of great interest to the field of SERS and catalysis. However, there is no direct evidence as to whether the target molecule and the target reactant molecule can actually reach the surface of the functional materials.

We agree with the reviewer that the chemical substances, such as TBA⁺ and CNT, have been studied previously in interfacial self-assembly. Indeed, this was mentioned in our manuscript mainly in the introduction section. We have added the two new papers suggested by the reviewer in our references since these are important work in the field, and modified the introduction section to improve its clarity. As the reviewer very accurately pointed out in the previous comment, the main novelty and step change provided by our approach is that we have rationally combined stabilizers and promoters in a novel way which allowed stable Pickering emulsions systems to be constructed without needing to modify the surface of the functional particles. As a result, this resolves the long-standing dilemma of choosing between stability and functionality in the synthesis of Pickering emulsions.

We are glad that the reviewer shares our views that the construction of “surface-accessible Pickering emulsions” is of great significance for applications, such as SERS and catalysis. We believe that the wide range of successful SERS and catalysis experiments shown in our work directly demonstrates that the target molecules are able to access the functional nanoparticles since these applications can only be achieved when the reactant or analyte is directly interacting with the surface of the functional material. Indeed, this is also consistent with our experimental data that show the same applications cannot be achieved when the functional particles are covered with strongly adsorbed modifiers. Interactions between the functional surface of the modifier-free emulsions can also be directly probed directly and in situ via SERS. For example, the Figure below compares the SERS signals acquired from the surface-accessible CNT-Au Pickering emulsions before and after being treated with 10⁻⁵ M of adenine. As shown, in spectrum i, the blank control spectrum is dominated by vibrational bands of citrate and Au-Cl which arise from the initial capping ligands adsorbed on the surface of the Au nanoparticles. When the emulsions are allowed to interact with adenine, the SERS signals of the capping ligands disappeared completely and are replaced by the vibrational bands of adenine which shows that the analyte molecules are interacting with the plasmonic surface by displacing the initial capping ligands. In contrast, the SERS signals of the Pickering emulsions constructed using modifiers remain unchanged even with the addition of higher concentrations of analytes, since the analyte molecules are unable to displace the strongly adsorbed modifiers to access the functional surface (as shown in Fig. 5b, spectra i-ii in the revised manuscript). We have added the Figure discussed above as Supplementary Fig. 14 in this revision. In addition, we have updated Fig. 5b and modified the main text accordingly. For convenience the changes are also shown below.

Supplementary Figure 14 | Probing the adsorption of adenine on modifier-free CNT-Au emulsions via SERS. SERS spectra obtained from CNT-Au emulsions before (ii) and after (i) interacting with 10⁻⁵ M of adenine.

Figure 5 | Applications of w/o CNT-Au Pickering emulsions in SERS. (b) SERS spectra obtained from 3-MPA-capped Au emulsions and CNT-Au emulsions treated with various concentrations of adenine.

“However, this potential application has barely been fulfilled since the strongly-adsorbed modifiers, which had been essential for constructing the plasmonic emulsions, prevent analyte molecules from interacting with the plasmonic nanosurface and accessing the enhancing hot spots (illustrated in the schematic of Fig. 5b). For example, Fig. 5b compares the SERS performance of Au emulsions formed using 3-MPA modifiers, against Au emulsions formed with promoters and CNT stabilizers in the detection of adenine. Even though adenine is a biomolecule which is known to adsorb spontaneously to the surface of Au nanoparticles, the adenine molecules were unable to displace the strongly adsorbed 3-MPA modifiers. As a result, no signals of the analyte could be observed from the modifier-capped emulsions when they were allowed to interact with 10⁻⁴ M of adenine (Fig. 5b, spectra *i-ii*). In sharp contrast, the surface of the Au nanoparticles in the CNT-Au emulsions only contained weakly adsorbed citrate and chloride capping ligands, which could be easily displaced by adenine (Supplementary Fig. 14). As a result, clear SERS signals of adenine could be obtained even when its concentration was 1000× lower (10⁻⁷ M) than the concentration used for modifier-capped emulsions (Fig. 5b, spectra *iii-vi*).”

3. In line 89-91, the first step to generating stable Pickering emulsions is to create favorable conditions which allow the charged nanoparticles to overcome interparticle electrostatic repulsion, it can be seen from Fig. 2B that CNT plays the role of overcoming interparticle electrostatic repulsion, but it is not explained in detail why CNT can overcome interparticle electrostatic repulsion. Please provide the necessary explanation.

We do not think that the CNTs (or more generally, the particle “stabilizers”) play a major role in helping the Au nanoparticle (or more generally, the “functional material”) overcome interparticle electrostatic repulsion. This is evident by the observation that self-assembly of the charged nanoparticles at the water-oil interface does not occur without promoters even if the stabilizer is present. Taking CNT-Au emulsions as an example, the CNT and Au nanoparticles are dispersed in the oil and water phase, respectively (as shown in Fig. 1a of the revised manuscript). Upon adding the promoter and shaking, the CNTs and Au nanoparticles migrate to the water-oil interface independently of each other. We note that without the promoters, the uncharged CNTs are able to assemble at the interface since this is energetically favourable but the charged Au nanoparticles cannot since the gain in electrostatic potential by packing the charged Au nanoparticles densely at the interface makes self-assembly energetically unfavourable. This effect is generally observed for all the particles used in this work i.e. only uncharged particles (or weakly charged, such as PVP-stabilized particles) can migrate spontaneously to the interface without promoters, while charged nanoparticles cannot assemble at the interface without promoters even when all the other ingredients are present in the system.

We think by “Fig. 2B” the reviewer is actually referring to Fig. 1b, since Fig. 2B is an optical image of the CNT-Au Pickering emulsions. The intention of Fig. 1b was to show that the particle layer at the interface in the Pickering emulsions consisted of both CNT and Au nanoparticles and that the CNTs provided “emulsion stabilization”, while promoters provided “charge screening”. From the reviewer’s comment, we realized that the original Figure did not clearly convey this message. To address the issues discussed above, and in particular, to clearly introduce the role of promoters and stabilizers, we have modified Fig. 1 and added corresponding discussions in the main text, which for convenience is shown below.

Figure 1 | The working principles of “stabilizers” and “promoters”. (a) Schematic illustration of the experimental process for generating carbon nanotube (CNT)-Au emulsions. (b)-(c) Schematic illustrations of the working principle of promoters. (d) Three-phase contact angle of CNTs, Au and CNT-Au nanoparticle layers measured for water-cyclohexane. (e)-(g) Photographs of CNT emulsions, Au films, and CNT-Au emulsions. (h)-(i) Schematic illustrations of the working principle of CNT stabilizers.

“Equation (1) shows that the adsorption of nanoparticles to the water-oil interface is generally a favourable process, provided that θ is not 0° or 180° . Indeed, spontaneous interfacial self-assembly was observed for the uncharged CNTs dispersed in oil, when shaken with water.

However, for electrostatically stabilized colloidal nanoparticles, the overall change in system energy due to self-assembly can no longer be described using just Equation (1), which is based solely on the reduction in interfacial energy. The additional gain in electrostatic potential from the nanoparticles packing closely at the interface must be considered, which can be expressed as below²⁶.

$$\mu_{coulomb,w} = \frac{z\pi R^3 \sigma^2}{8\epsilon_w \epsilon_0} \left(3\sqrt{y} + \frac{2R}{L_D} \right) e^{L_D \sqrt{y}} \quad (2)$$

$$\mu_{coulomb,o} = \frac{3z\pi R^3 \alpha^2 \sigma^2 \sqrt{y}}{8\epsilon_{oil} \epsilon_0} \quad (3)$$

More specifically, Equations (2)-(3) show the electrostatic potential between two similarly charged spherical nanoparticles within a hexagonally packed nanoparticle-lattice on either side of the interface (Supplementary Fig. 1), where $\mu_{coulomb,w}$ and $\mu_{coulomb,o}$ are the electrostatic potential between the parts of the nanoparticles immersed in the water and oil phases, respectively, z is the number of nearest neighbouring particles, y is the fraction of interface covered by nanoparticles versus the maximum fraction of interface that could be covered by the nanoparticles, σ is the particle charge density, ϵ_{oil} and ϵ_w are the dielectric constants of the oil and water phase, and ϵ_0 is the permittivity of vacuum. Since the charged species on the surface of the nanoparticles are inherently hydrophilic, it is unlikely that they can be completely transferred across the interface into the oil phase. Additionally, some of the charged species which remain on the surface of the particles could recombine with their counter-ion upon entering the oil phase. Therefore, a scaling factor α , which ranges between 0 and 1, is introduced in Equation 3.

In practice, this means that electrostatically stabilized colloidal nanoparticles, such as the Au nanoparticles used here, do not migrate spontaneously to the interface when the colloid is simply shaken with an immiscible oil (Fig. 1b and Supplementary Fig. 2). Therefore, the key to inducing self-assembly of charged colloidal nanoparticles at water-oil interfaces is to lower the electrostatic repulsion between adjacent particles at both sides of the interface. As discussed above, the conventional approach to achieve this is through the addition of organic modifiers, such as thiols, or co-solvents, such as ethanol, but modifiers passivate the surface of the functional nanoparticles while co-solvents destabilize emulsion systems. Previously, we and others have shown that an alternative approach to induce self-assembly of charged colloidal nanoparticles is to use promoters, such as tetrabutylammonium (TBA⁺) nitrate, which are lipophilic or amphiphilic organo-electrolytes that carry an opposite charge to the nanoparticles²⁷⁻²⁹. As shown in Fig. 1c, the promoters along with the counterions, such as Na⁺, which are initially present in the colloidal solution sit between the charged nanoparticles at each side of the interface to provide charge-screening which helps overcome interparticle electrostatic repulsion. It is useful to note that this method is also fundamentally different from previous reports which use simple hydrophilic salts, such as NaCl, to induce partial aggregation of the colloid and therefore lower interparticle electrostatic repulsion at the interface³⁰. Here, the addition of promoters does not lead to any observable aggregation as shown by UV-vis spectroscopy (Supplementary Fig. 3). Therefore, using promoters allows the charged colloidal nanoparticles to be assembled into densely packed superstructures without passivating their surface with strongly adsorbed modifiers or inducing undesired aggregation. In principle, the minimum amount of promoter required to effectively provide charge screening will differ depending on the surface-charge of the colloid used for self-assembly. However, this is not an issue in practice since a reasonable excess of promoter can be used routinely without perturbing the emulsion system. We have found that a promoter concentration of ca. 3×10^{-5} M (per volume of colloid) was adequate for inducing self-assembly of highly charged colloids with zeta potential ca. -40 mV²⁸ (Supplementary Fig. 4). Based on the above, a promoter concentration of 2.4×10^{-3} M was consistently used for all emulsion systems in this work to mitigate interparticle electrostatic repulsion during self-assembly.”

In addition, we have also added new experimental data in the Supplementary Information to highlight the crucialness of promoters for overcoming interparticle electrostatic repulsion. More specifically, we showed that without the addition of promoters, the charged colloidal nanoparticles remain in the bulk aqueous phase after shaking, and as a result, no emulsions are formed (Supplementary Fig. 2). The addition of promoters does not lead to unwanted particle aggregation (Supplementary Fig. 3) but allows the charged particles to migrate to the interface.

This is not only evident by the successful formation of stable Pickering emulsions, but also by the observations that the aqueous phase becomes clear and colourless after self-assembly (Supplementary Fig.4), which indicates that the nanoparticles have migrated to the water-oil interface. This is also supported by UV-vis and SERS analysis of the emulsion system and parent colloids which show that the number of plasmonic nanoparticles remaining in the bulk aqueous phase after promoter induced self-assembly is negligible (Supplementary Fig. 4 and 10). For convenience the newly added Supplementary Figures are also shown below.

Supplementary Figure 1 | Schematic illustration of a densely packed hexagonal array of spherical nanoparticles. R corresponds to the radius of the particles, x corresponds to the surface-to-surface distance between two adjacent particles. Figure reproduced from Ref 8.

Supplementary Figure 2 | Demonstration that promoters are essential for self-assembly. Optical images showing a sample of citrate-reduced Ag nanoparticles and SiO₂ nanoparticles before and after being shaken with dichloromethane without any promoters. The sample remains unchanged, and the nanoparticles do not migrate to the water-oil interface. SiO₂ stabilizers were selected as an example since they are charged and dispersed in the aqueous phase.

Supplementary Figure 3 | Probing the influence of tetrabutylammonium (TBA) salt on colloid stability. UV-vis spectra of citrate-reduced Ag and Au colloid with and without the addition of TBA promoters. The concentration of promoters was the same as that used for the synthesis of Pickering emulsions.

Supplementary Figure 4 | Demonstration of the migration of nanoparticles to the water-oil interface in promoter-assisted self-assembly. (a)-(b) Optical images showing a sample of citrate-reduced Ag nanoparticles and SiO₂ nanoparticles before and after being shaken with dichloromethane and promoters. (c) UV-vis spectra of the aqueous colloid phase before and after self-assembly showing that a negligible number of nanoparticles remain in the bulk aqueous phase after the formation of Pickering emulsions.

4. In Fig.2F, the stability of the CNT-Au emulsions over time is considered. It is well known that in substrates used for SERS detection, the stability of the substrate is important for the detection effect. And whether the emulsion formed by CNT-Au has an advantage in terms of stability compared to the common surface-modified emulsion, such as 3-MPA-capped Au emulsions.

We did not show the time dependent stability of the CNT-Au emulsions as SERS substrates compared to the MPA-capped emulsions since the modified emulsions cannot act as an effective substrate for common analytes even when freshly prepared due to its surface being covered with strongly adsorbed modifiers. Inspired by the reviewer's comment, in this revision, we have performed additional experiments to showcase the stability of the functional properties of the surface-accessible emulsions over time. More specifically, using the catalytic reduction of 4-nitrophenol as an example, we showed that the surface-accessible emulsions fully retain their catalytic activity even after being stored at room temperature for one month. The results

of this experiment have been added to the revised main text and Fig. 6f and the Supplementary Information as Supplementary Fig. S23. For convenience the new Figures and revised main text are also presented below.

Figure 6 | Applications of o/w SiO₂-Ag Pickering emulsions in catalysis. (f) Plot showing the signal intensity of 4-NP measured by UV-vis spectroscopy at different points of catalytic reduction using fresh SiO₂-Ag Pickering emulsions and SiO₂-Ag Pickering emulsions that have been stored for 1 month as the catalyst. Inset compares the conversion rate of 4-NP using fresh and stored SiO₂-Ag emulsions.

Supplementary Figure 23 | Comparing the catalytic activity of fresh and stored SiO₂-Ag emulsions. (a) Optical image of a typical batch of freshly prepared SiO₂-Ag emulsions. (b)-(c) UV-vis spectra of 4-NP measured at different stages of the catalytic reduction using freshly prepared SiO₂-Ag emulsions as the catalyst, and the Ln(A/A₀) of 4-NP measured by UV-vis spectroscopy plotted against time. (d) Optical image of a typical batch of SiO₂-Ag emulsions that had been stored at room temperature for 30 days. (e)-(f) UV-vis spectra of 4-NP measured at different stages of the catalytic reduction using SiO₂-Ag emulsions that had been stored for 30 days as the catalyst, and the Ln(A/A₀) of 4-NP measured by UV-vis spectroscopy plotted against time. Panels (b) and (c) are also displayed in Fig. 6 of the main text. Panel (b) is also displayed in Supplementary Fig. 24, 25 and 26.

“...In addition to the superior catalytic activity, the modifier free emulsions also showed excellent stability over time. As shown in Fig. 6f and Supplementary Fig. 23, the SiO₂-Ag emulsions fully retained their catalytic activity even when left at room temperature for 1 month.

Finally, the generality of our co-assembly approach and stability of the product emulsions means that the composition of the emulsions and the reaction conditions can be readily adjusted to suit specific catalytic reactions...”

5. The role of promoters, such as tetrabutylammonium (TBA⁺) nitrate, is not clear. Not only is there a lack of control experiments for the promoters, but the G and H images in Figure 2 are very blurred. It is not possible to distinguish whether the nanoparticles (functional materials) are formed directly by adsorption onto CNTs or by forming nano islands and thus assembling into Pickering emulsions. Besides, there is also a lack of effective control experiments regarding SERS, and catalytic application-related experiments.

Following the reviewer’s suggestion, in this revision we have made significant modifications to both the main text and Supplementary Information by adding a series of new experimental data and discussions surrounding the working mechanism of the promoters, the in situ and ex situ characterization of the nanoparticle layers at the interface, and the optimization of SERS and catalytic applications.

To better illustrate and discuss the working mechanism of the promoters, we have updated the schematic in Fig. 1 of the main text. We have also added experiments to systematically demonstrate the importance of the promoters in self-assembly of charged colloidal nanoparticles, which is shown in Supplementary Fig. 2-4 in the revision. Correspondingly, we have added new discussions in the main text. The changes above can be found in detail in the reply to Question 3 by Reviewer 4.

To improve the quality of our electron microscopy analysis, we have updated the SEM of the CNT-Au emulsions with newly obtained TEM images, which showed more clearly that “the CNTs were always spread out evenly as an entangled mesh while the Au nanoparticles occupied the gaps in the form of islands”. For convenience the updated Fig. 3h in the main text is also shown below.

Figure 3 | Microscopic characterization of CNT-Au emulsions. (h) Transmission microscopy image of a typical area in a CNT-Au nanoparticle layer formed using 0.05 mg mL⁻¹ of CNTs.

The catalytic and SERS applications were only meant to demonstrate how removing the modifiers leads to significantly improved functionality. This was why we did not perform extensive optimization of the catalytic system in our initial submission, and only showed comparisons between the catalytic activity of the modified and un-modified Pickering emulsions under the exact same experimental conditions.

In this revision we have added experiments to show that the catalytic activity of the surface-accessible Pickering emulsions formed via our new approach can be readily optimized. More specifically, we have performed experiments to study the stability of the emulsions, and to optimize the reaction temperature, reactant concentration, stabilizer concentration and metal catalysts type. In general, our experimental data was consistent with the findings in literature for the catalytic reduction of 4-nitrophenol. More importantly, our new experimental data demonstrates that even without extensive system optimization, the modifier-free emulsions shown in our work already exhibit excellent functionalities. The data for these new experiments are presented in Fig 6, and Supplementary Fig. 21-27 of the revised manuscript.

Similar to the case with the catalysis demonstrations, we did not include extensive optimizations of the SERS studies since the focus was to show that removing the modifiers led to significant

improvements in the functionality of the Pickering emulsions. Using adenine and crystal violet as the model analytes, in this revision we have added concentration-dependent SERS data obtained using the modifier-free Pickering emulsions as SERS substrates. In general, the surface-accessible Pickering emulsions showed high plasmonic activities which allowed SERS quantitation to be achieved. As shown in Supplementary Fig. 15, even with simple citrate-reduced Au nanoparticles as the plasmonic component, the limit of detection for adenine and crystal violet could reach 10^{-7} M and 5×10^{-7} M, respectively. The linear quantitation range was determined to be 10^{-4} - 10^{-7} M and 10^{-5} - 5×10^{-7} M for adenine and crystal violet, respectively. For convenience the series of updated Figures are also shown below.

Supplementary Figure 15 | SERS Quantitation with CNT-Au emulsions. (a)-(b) SERS quantitation of crystal violet and adenine using CNT-Au emulsions as the enhancing substrate. Insets show the calibration curves obtained by plotting $\text{Log}(C_{\text{analyte}})$ versus $\text{Log}(I_{\text{analyte}})$.

Figure 6 | Applications of o/w SiO_2 -Ag Pickering emulsions in catalysis. (a) Schematic illustrations of a modifier-free o/w SiO_2 -metal emulsion droplet acting as an interfacial catalyst for the reduction of 4-nitrophenol (4-NP) to 4-aminophenol (4-AP). (b) Optical images of a Ag emulsion sample formed with 3-MPA modifiers before and a few minutes after being treated with NaBH_4 solution. Also shown is the schematic illustrations of the 3-MPA modifiers being replaced

by H^- in the $NaBH_4$ solution leading to emulsion destabilization. (c) Optical images of a SiO_2 -Ag emulsion sample before and after 20 minutes of catalytic reaction. Also shown is the schematic illustrations of the surface-accessible Ag nanoparticles in modifier-free emulsions acting as stable catalysts. (d) UV-vis spectra of 4-NP measured at different stages of its catalytic reduction using SiO_2 -Ag Pickering emulsions as the catalyst. (e) Semi-log plot of $\ln(A/A_0)$ against time for 4-NP. (f) Plot showing the absorbance of 4-NP measured by UV-vis spectroscopy at different points of catalytic reduction using fresh SiO_2 -Ag Pickering emulsions and SiO_2 -Ag Pickering emulsions that have been stored for 1 month as the catalyst. Inset compares the overall conversion of 4-NP using fresh and stored SiO_2 -Ag emulsions.

Supplementary Figure 20 | SiO_2 -noble metal emulsions for catalysis. (a)-(c) Optical images of typical batches of SiO_2 -Au, Ag and Pt Pickering emulsions, respectively. (d)-(f) SEM characterizations of SiO_2 -Au, Ag and Pt Pickering emulsions.

Supplementary Figure 21 | Probing the stability of SiO_2 -Ag emulsions after acting as interfacial catalysts. Optical microscopy images of a typical batch of SiO_2 -Ag emulsion before and after catalytic reduction of 4-NP. The plot shows the average diameter and standard deviation of the emulsion droplets measured from the two images.

Supplementary Figure 23 | Comparing the catalytic activity of fresh and stored $\text{SiO}_2\text{-Ag}$ emulsions. (a) Optical image of a typical batch of freshly prepared $\text{SiO}_2\text{-Ag}$ emulsions. (b)-(c) UV-vis spectra of 4-NP measured at different stages of the catalytic reduction using freshly prepared $\text{SiO}_2\text{-Ag}$ emulsions as the catalyst, and the $\text{Ln}(A/A_0)$ of 4-NP measured by UV-vis spectroscopy plotted against time. (d) Optical image of a typical batch of $\text{SiO}_2\text{-Ag}$ emulsions that had been stored at room temperature for 30 days. (e)-(f) UV-vis spectra of 4-NP measured at different stages of the catalytic reduction using $\text{SiO}_2\text{-Ag}$ emulsions that had been stored for 30 days as the catalyst, and the $\text{Ln}(A/A_0)$ of 4-NP measured by UV-vis spectroscopy plotted against time. Panels (b) and (c) are also displayed in Fig. 6 of the main text. Panel (b) is also displayed in Supplementary Fig. 24, 25 and 26.

Supplementary Figure 24 | Effect of temperature on the catalytic performance of $\text{SiO}_2\text{-Ag}$ emulsions. (a)-(d) UV-vis spectra of 4-NP measured at different stages of the catalytic reduction using $\text{SiO}_2\text{-Ag}$ emulsions as the catalyst at 15 °C and 35 °C, respectively, and the $\text{Ln}(A/A_0)$ of 4-NP measured by UV-vis spectroscopy plotted against time. Panels (b) and (c) are also displayed in Fig. 6 of the main text. Panel (b) is also displayed in Supplementary Fig. 23, 25 and 26.

Supplementary Figure 25 | Effect of reactant concentrations on the catalytic performance of $\text{SiO}_2\text{-Ag}$ emulsions. (a)-(c) UV-vis spectra of 4-NP measured at different stages of the catalytic reduction using $\text{SiO}_2\text{-Ag}$ Pickering emulsions as the catalyst at various NaBH_4 concentrations. Panel (b) is also displayed in Fig. 6 of the main text and Supplementary Fig. 23, 24 and 26.

Supplementary Figure 26 | Effect of stabilizer concentrations on the catalytic performance of $\text{SiO}_2\text{-Ag}$ emulsions. (a)-(c) UV-vis spectra of 4-NP measured at different stages of the catalytic reduction using $\text{SiO}_2\text{-Ag}$ Pickering emulsions as the catalyst at various SiO_2 concentrations. Panel (b) is also displayed in Fig. 6 of the main text and Supplementary Fig. 23, 24 and 25.

Supplementary Figure 27 | Effect of catalyst composition on the catalytic performance of SiO₂-noble metal emulsions. (a)-(c) UV-vis spectra of 4-NP measured at different stages of the catalytic reduction using SiO₂-noble metal Pickering emulsions containing different types of metal nanoparticles. The metal catalysts weight content for Au and Pt were 1.55×10^{-3} and 2.22×10^{-3} (wt. %), respectively. Panel (c) is also displayed in Fig. 6 of the main text and Supplementary Fig. 23, 24 and 25.

- In Fig. 2F, the addition of CNTs with concentrations of 0.025 mg/mL and 0.05 mg/mL have about a two-fold difference in the size of the emulsion. Please refine the concentration gradient.

We have refined the concentration gradient in this revision. The revised Figure is presented below.

Figure 3 | Microscopic characterization of CNT-Au emulsions. (a-f) Optical images of CNT-Au emulsions formed with 0.025, 0.0375, 0.05, 0.1, 0.15, and 0.2 mg mL⁻¹ of CNTs, respectively. The scale bars correspond to 1 mm. (g) Plot comparing the average size and stability of the emulsions versus the amount of CNTs used.

- Figure 4b compares the detection sensitivity of 3-MPA-capped Au emulsions and Au -CNT emulsions as SERS substrates. The sulfhydryl molecules in the 3-MPA-capped Au emulsions act as modified molecules that replace the citrate barrier analytes on the AuNP surface and thus affect the detection sensitivity, whereas CNT does not act as a barrier molecule to reach the AuNP surface but does not replace the citrate on the AuNP surface, and CNT only stabilizes to some extent. This method still does not solve the problem that AuNP surfactant affects the sensitivity of SERS detection.

The reviewer summarized our work very accurately, the involvement of CNTs, or more generally, particle stabilizers remove the requirement for modifying the surface of the functional material (in this case, Au nanoparticles) with modifiers. This paves the way for surface-related applications.

We are not sure what the reviewer meant by “AuNP surfactant”. We have assumed that the reviewer is referring to surfactants, such as CTAB, which are often used as growth directing agents in the synthesis of plasmonic nanoparticles and end up affecting the functionalities of the product particles. In general, we agree with the reviewer that our work here is only one step towards the synthesis of surface-accessible nanomaterials, and cannot be used to prevent the use of surfactants or modifiers during the synthesis of the functional nanoparticles. Here, the significance of work is that it demonstrates the importance of surface-accessibility in nanotechnology, and paves the way for the synthesis of surface-accessible Pickering emulsions with the surfactant-free nanoparticles currently available in literature. We believe that this will inspire new research surrounding surface-accessible nanomaterials, including the development of surfactant-free colloidal synthesis. Within this context, the self-assembly approach demonstrated in our work will continue to have a positive impact since it will provide a method for constructing the ever-expanding types of surfactant-free nanoparticles into functional assemblies while preserving their surface accessibility. More generally, the self-assembly approach demonstrated in this work is also useful for the wide range of surfactant-modified nanoparticles available in literature (for example the PVP-Ag nanocubes demonstrated in this work), since it removes the need for finding a suitable modifier and modifying the surface of these nanoparticles with even more strongly adsorbed molecular modifiers.

If by “surfactant”, the reviewer is referring to citrate, we would respectfully argue that capping ligands, such as citrate and chloride, are much more weakly adsorbed than common thiol modifiers, which make the surface of the nanoparticles much more accessible, and results in significantly enhanced functionalities. This was demonstrated in this work with SERS and catalysis, and in our previous work (for example, *Chem* 2022, 8, 2514; *JACS Au* 2022, 2, 178; *Angew. Chem. Int. Ed.* 2019, 131, 19230; *Nano Lett.* 2016, 16, 5255).

We have modified the conclusions of our manuscript in this revision to include the discussions above.

8. In addition to this, the paper also contains some obvious writing errors. For example, the phrase "oil-in-water" in line 107 of the text is not consistent with the presentation of the paper.

We have now corrected this in the current revision.

Reviewer comments, further round review –

Reviewer #1 (Remarks to the Author):

The authors have answered my concerns as required, I think it can be accepted as current form.

Reviewer #2 (Remarks to the Author):

The revisions suggested from my side has been well addressed by the authors. I recommend the manuscript to be accepted for publication.

Reviewer #3 (Remarks to the Author):

The authors have appropriately addressed the previous comments and made corresponding revisions to the manuscript. I do not have further questions. I am happy to recommend acceptance of this paper for publication without further changes.

Reviewer #4 (Remarks to the Author):

It is suitable for publication now.